TECHNIQUES FOR PHYSIOLOGY

# Development and characterisation of a rat model that exhibits both metabolic dysfunction and neurodegeneration seen in type 2 diabetes

Katherine Southam[1,3], Chantal de Sousa[1], Abraham Daniel[1,2] , Bruce V. Taylor[3], Lisa Foa[1,4,*] and Dino Premilovac[1,*]

[1] *Tasmanian School of Medicine, College of Health and Medicine, University of Tasmania, Hobart, TAS, Australia*
[2] *School of Pharmacy and Pharmacology, College of Health and Medicine, University of Tasmania, Hobart, TAS, Australia*
[3] *Menzies Institute for Medical Research, College of Health and Medicine, University of Tasmania, Hobart, TAS, Australia*
[4] *School of Psychological Sciences, College of Health and Medicine, University of Tasmania, Hobart, TAS, Australia*

Edited by: Kim Barrett & Maria Chondronikola

The peer review history is available in the Supporting information section of this article (https://doi.org/10.1113/JP282454#support-information-section).

**The Journal of Physiology**

Katherine Southam completed her PhD with a primary focus on motor neuron degeneration in amyotrophic lateral sclerosis, including the development of novel cell culture techniques to investigate neuron–glia interactions. She has subsequently expanded her interest in neuronal degeneration during Alzheimer's disease and type 2 diabetes and, more recently, has investigated the effects of environmental pollution on brain development. Katherine is based at the University of Tasmania.

*L. Foa and D. Premilovac contributed equally to this work.

**Abstract** Accurate modelling type 2 diabetes and diabetic complications in rodents has proven a challenge, largely as a result of the long-time course of disease development in humans. In the present study, we aimed to develop and comprehensively characterise a new rodent model of type 2 diabetes. To do this, we fed Sprague–Dawley rats a high fat/high sugar diet (HFD) to induce obesity and dyslipidaemia. After 3 weeks, we s.c. implanted osmotic mini pumps to enable a 14 day, slow infusion of streptozotocin (STZ; lower dose = 100 mg kg$^{-1}$; higher dose = 120 mg kg$^{-1}$) to dose-dependently reduce pancreatic beta cell mass. After removing the mini pumps, we monitored animals for 4 months using a battery of tests to assess both metabolic and neurodegenerative changes across time. Our data demonstrate the combination of the HFD and lower dose STZ leads to induction of early-stage type 2 diabetes defined by moderate hyperglycaemia, hyperinsulinaemia and impaired glucose tolerance, at the same time as the retention of an obese phenotype. By contrast, combining the HFD and higher dose STZ leads to induction of later-stage type 2 diabetes defined by frank hyperglycaemia, hypoinsulinaemia (but not insulin depletion) and severely impaired glucose tolerance, at the same time as retaining an obese phenotype. Regardless of dose of STZ (and level of hyperglycaemia), all diabetic rats exhibited signs of peripheral neurodegeneration in the skin and muscle. Thus, this model recapitulates many of the complex metabolic disturbances seen in type 2 diabetes and provides an excellent platform for investigating the pathophysiological mechanisms that lead to diabetic complications such as peripheral neuropathy.

(Received 12 October 2021; accepted after revision 2 February 2022; first published online 6 February 2022)

**Corresponding author** D. Premilovac: Tasmanian School of Medicine, College of Health and Medicine, University of Tasmania, Hobart, TAS 7000, Australia. Email: dino.premilovac@utas.edu.au

**Abstract figure legend** The combination of a long-term high fat diet and 2 week osmotic mini pump infusion of streptozotocin (STZ) enables induction of different stages of type 2 diabetes in rats. This method leads to sustained obesity, dyspidaemia and moderate hyperglycaemia, as well as either hyperinsulinaemia (100 gm kg$^{-1}$ STZ) or hypoinsulinaemia (120 mg kg$^{-1}$ STZ), depending on the dose of STZ utilised. Over the course of ∼4 months, these animals develop signs of peripheral nerve degeneration in skin and muscle as seen in type 2 diabetes.

**Key points**

- Type 2 diabetes is a major health concern and markedly increases risk cardiovascular and neurodegenerative diseases.
- Accurate modelling of type 2 diabetes is a major challenge and has impeded our ability to understand the mechanisms that contribute to complications of type 2 diabetes.
- We have developed a method of inducing different stages of type 2 diabetes using a high fat/high sugar diet and 14 day infusion of streptozotocin to dose-dependently destroy pancreatic beta cell mass.
- Over 4 months, we comprehensively characterised these animals and confirmed that they develop sustained metabolic dysfunction and progressive peripheral neurodegeneration as seen in type 2 diabetes.
- This new model will improve our ability to investigate the pathophysiological mechanisms that link type 2 diabetes with complications such as neurodegeneration.

## Introduction

Type 2 diabetes is the most prevalent form of diabetes worldwide, accounting for ∼90% of cases. The incidence of type 2 diabetes is increasing, with conservative estimates suggesting that around 642 million people worldwide will be living with the condition by 2040 (Guariguata *et al.* 2014; IDF, 2015). Management of type 2 diabetes and associated cardiovascular and neurological complications places a large burden on the health system in both developing and developed countries as a result of the long-term management of hyperglycaemia and complications (Guariguata *et al.* 2014; IDF, 2015). This is especially concerning in developing countries where an estimated 50% of people with type 2 diabetes lack access to appropriate management for their condition and have

a greater probability of developing neurological disorders such as peripheral neuropathy and retinopathy, increasing their morbidity and mortality (Reaven, 1988; Brownlee, 2001; Kahn *et al.* 2006; Chen *et al.* 2011; Guariguata *et al.* 2014; IDF, 2015). Beyond neuropathies, over the last 10–20 years, type 2 diabetes has been described as a major risk factor for the development of brain disorders such as dementia and Alzheimer's disease (Biessels & Reagan, 2015). Although a wealth of research has endeavoured to reveal how type 2 diabetes develops and progresses to cause neurological complications (Biessels & Reagan, 2015), the difficulty in unravelling the complex and multiple pathophysiological processes present in people with type 2 diabetes has impeded progress. Animal models provide an opportunity to inform many of these pathophysiological processes, but none are currently capable of reproducing the human form of the disease, primarily as a result of the ∼10 year preclinical stage of type 2 diabetes in humans.

It is well accepted that people progress through two distinct stages during development of type 2 diabetes. In most cases, the first stage is driven by obesity-associated insulin resistance impacting the function of multiple organs such as the liver and skeletal muscles (Reaven, 1988; Kahn *et al.* 2006). In this stage, normal circulating glucose concentrations are maintained by a compensatory increase in pancreatic insulin production (Reaven, 1988; Kahn *et al.* 2006; Premilovac *et al.* 2013). As the background insulin resistance increases, the need for increased insulin production compromises pancreatic beta cell function. Eventually, beta cells become exhausted and hyperglycaemia manifests as a result of impaired insulin production and secretion, with this comprising the second stage of type 2 diabetes. Historically, hyperglycaemia and subsequent glucose toxicity has been championed as the primary driver of cardiovascular and neurological complications and increased morbidity as a result of type 2 diabetes (Brownlee, 2001). Recent work has identified factors such as multi-organ insulin resistance and dyslipidaemia as additional co-conspirators (Di Pino & DeFronzo, 2019). However, the exact interplay between hyperglycaemia, insulin resistance and dyslipidaemia, as well as how these lead to diabetic complications, has been difficult to untangle. This gap in the literature has been difficult to fill partly because of inadequate animal models of type 2 diabetes.

Apart from transgenic rodents, which do not replicate the aetiology of the human condition, the two-stage progression to type 2 diabetes has been modelled in experimental animals using a combination of high fat diet (HFD) and streptozotocin (STZ) (Oakes *et al.* 1997; Reed *et al.* 2000; Srinivasan *et al.* 2005; Srinivasan & Ramarao, 2007; Anderson *et al.* 2014; Podell *et al.* 2017). In the HFD + STZ model, 2–10 weeks of HFD feeding is used to induce obesity-associated insulin resistance (Srinivasan *et al.* 2005; Premilovac *et al.* 2017) after which animals are typically injected with STZ, a pancreatic beta cell toxin, to quickly reduce or deplete beta cell insulin production. Although this is an extensively used model, there is a lack of consistency in the literature regarding STZ dosage, frequency and the route of STZ administration, resulting in a high degree of variability in the phenotype of the animals (Skovsø, 2014). Most, but not all studies, using injectable STZ (ɪ.ᴘ. or ɪ.ᴠ.) use high STZ doses and report a rapid transition from obesity and normoglycaemia to overt hyperglycaemia and cachexia, more akin to type 1 diabetes (Stranahan *et al.* 2008; Kleinridders *et al.* 2014). Some utilise multiple low-dose STZ injections to induce type 2 diabetes, but this is reported to be transiently effective where beta cells appear to be able to compensate or regenerate and normalise glucose levels (Zhang *et al.* 2009; Skovsø, 2014). Thus, injectable STZ models do not appropriately reproduce the pathophysiological changes seen in humans, with these being insulin resistance and modest hyperglycaemia, at the same time as the retention of an obese and hyperlipidaemic phenotype. Beyond injection methods, we have recently demonstrated that administration of STZ using osmotic mini pumps provides exquisite control over the level of resultant hyperglycaemia at the same time as maintaining an obese phenotype in the subsequent 2 weeks (Premilovac *et al.* 2017). The present study aimed to determine whether this new method could be used to induce different stages of type 2 diabetes in rats and establish whether these animals develop associated neurological complications seen in humans such as neuropathy and retinopathy.

## Methods

### Ethical approval

All experimental procedures were approved by the University of Tasmania Animal Ethics Committee (application id: A0016524) and were performed in accordance with the Australian Code for the Care and Use of Animals for Scientific Purposes – 2013, 8th Edition. Male Sprague–Dawley rats aged 14–16 weeks were obtained from the University of Tasmania animal facility and allowed to acclimatise for 7 days upon arrival. Rats were randomised and housed in groups of three under a 12:12 h light/dark photocycle at 21 ± 2°C. All animals were provided with water and chow *ad libitum* for throughout the study. Apart from control diet (CD) fed rats, all other groups were provided with a HFD (23% fat by weight; simple carbohydrate replacement; Specialty Feeds, Glen Forrest WA, Australia) *ad libitum* throughout the study (Fig. 1). In total, 36 rats were used in the present study.

## STZ dosage and osmotic mini pump implantation

The STZ (Sigma-Aldrich, St Louis, MO, USA) doses used in the present study were 100 mg kg$^{-1}$ (termed 'lower STZ') and 120 mg kg$^{-1}$ (termed 'higher STZ') and were chosen based on our previous work demonstrating different effects on glucose levels over 2 weeks (Premilovac *et al.* 2017; Daniel *et al.* 2021). STZ was prepared in citrate buffered saline (0.1 mmol L$^{-1}$; pH 4.4) and a total volume of 2 mL was loaded into mini pumps 20–30 min prior to surgical implant. Vehicle treated rats received mini pumps containing citrate buffered saline (pH 4.4; 2 mL). To control for any diet-specific effects, two additional groups of rats were maintained on CD or HFD only for the duration of the study (Fig. 1).

After 3 weeks of HFD, STZ was infused at a constant rate (5 $\mu$L h$^{-1}$) over a 14 day period using an s.c. implanted osmotic mini pump (Alzet Model 2ML2; Durect Corporation, Cupertino, CA, USA). Rat anaesthesia was maintained using isoflurane (2–3%) throughout the $\sim$15 min surgical procedure with monitoring of anaesthesia depth using respiratory rate/depth and response to tail and foot pinch. Following sterile preparation of the surgical site, a 1.5 cm incision (perpendicular to the spine) was made through the skin on the dorsum of the animals at the lumbar level of the spinal cord. Blunt dissection was used to separate connective tissue between the skin and underlying muscle layers to allow for insertion of the osmotic mini pump. Vehicle or STZ-loaded mini pumps were inserted and positioned inside the cavity between the scapulae so that delivery of solution occurred away from the surgical site.

The incision site was sealed using dissolvable sutures and staples and animals allowed to recover. To provide pain relief after surgery, rats were s.c. injected with meloxicam (0.05 mg kg$^{-1}$; Ilium; Troy Laboratories, Glendenning, NSW, Australia).

## Assessment of metabolic alterations

Food intake, water intake, body weight and non-fasting blood glucose concentrations were assessed every weekly for the duration of the 21 week intervention (Fig. 1). Non-fasting glucose concentrations were determined using a hand-held glucometer (Accu-Chek Performa; Roche Diagnostics, North Ryde, NSW, Australia) by making a small incision to the tip of the tail to access blood. Non-fasting glucose concentrations were assessed daily during the 14 day delivery of STZ to monitor development of hyperglycaemia during the STZ infusion. Glucose tolerance tests were performed at weeks 1, 3, 6, 10, 14 and 18 to monitor changes in glucose handling across the 21 week protocol. Because all glucose tolerance tests were performed after an overnight fast, plasma was collected before each test to determine fasting glucose, insulin (Rat Insulin ELISA; Mercodia, Uppsala, Sweden) and non-esterified fatty acid concentrations (NEFA; NEFA-C; Wako Pure Chemical Industries, Osaka, Japan) across time in each group.

All glucose tolerance tests were performed on overnight fasted rats. Briefly, animals received an i.p. injection of glucose (1 g kg$^{-1}$). Blood was accessed from the tip of the tail and glucose concentrations were measured

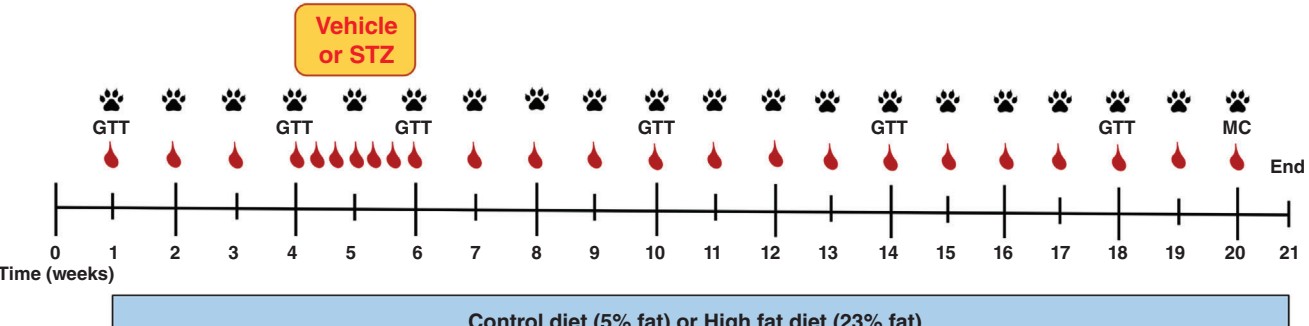

**Figure 1. Experimental timeline**
All rats were entered into a 21 week intervention and were provided with either control (5% fat w/w) or a HFD (23% fat w/w) *ad libitum*. All rats on the HFD had osmotic minipumps implanted s.c. to deliver vehicle (HFD + sham), 100 mg kg$^{-1}$ STZ (HFD + lower STZ) or 120 mg kg$^{-1}$ STZ (HFD + higher STZ) between weeks 4 and 6. During this 2 week period, non-fasting blood glucose (🩸) was monitored daily to track development of hyperglycaemia. Body weight and non-fasting glucose concentrations (🩸) were assessed once per week in all rats for the duration of the experiment. GTT (1 g of glucose per kg body weight) were performed in weeks 0, 4, 6, 10, 14 and 18. A meal challenge (MC) test was performed in week 20 by providing overnight fasted rats with their respective diet to eat *ad libitum* for 30 min. Prior to the experimental protocol, all rats were acclimatised to the plantar aesthesiometer set-up to enable weekly assessment of paw withdrawal threshold (🐾) once per week across the 21 week time course. All animals were killed in week 21 and tissues were collected for histological analysis. [Colour figure can be viewed at wileyonlinelibrary.com]

at 0, 10, 20, 30, 45, 60, 75, 90, 105 and 120 min after glucose injection. During each glucose tolerance test, larger volumes of blood (4–5 drops) were collected at 0, 10, 20 and 60 min after glucose injection in pre-heparinised tubes, centrifuged and plasma stored at −20°C. Plasma insulin concentrations during the glucose tolerance test were determined at a later date as above (Mercodia).

During week 20, rats underwent a meal challenge to assess post-prandial blood glucose concentrations between groups. In this metabolic test, rats were fasted overnight and provided with their allocated diet at 09.00 h the next morning. The diet was placed on the floor of the cage so that rats could consume the food *ad libitum* for 30 min. After 30 min, the left-over food was taken away and blood glucose concentrations assessed as described above. In the meal challenge, blood glucose levels were assessed at 0, 30, 45, 60, 75, 90, 105 and 120 min.

### Behavioural assessment for peripheral neuropathy

A Dynamic Plantar Aesthesiometer (Ugo Basile, Gemonio, Italy) was used to assess touch sensitivity on the rear plantar surface of each rat by monitoring time and force at voluntary paw withdrawal. Briefly, animals underwent a 2 week acclimatized phase to the equipment before the start of the dietary intervention in week 1. The acclimatization phase and in-experiment testing involved placing the rats in small, opaque plastic cages located on a perforated platform. After 5 min, a movable force actuator with filament (Von Frey–type 0.5 mm rigid filament) was positioned below the plantar surface of the animal and the apparatus initiated. The filament applied increasing force (0–50 g) to the plantar surface over a 20 s assessment period. The time/force taken for the rat to voluntarily withdraw the rear paw was automatically recorded by the apparatus and represents the withdrawal threshold. Touch sensitivity was assessed by taking three readings for each animal and calculating the mean for each rat, once per week, for the duration of the experimental protocol (Fig. 1).

### Immunohistochemistry

The pancreas, eyes, hindleg plantar skin and tibialis muscle of each animal were immediately excised after death via $CO_2$ and fixed overnight using neutral buffered formalin (4% paraformaldehyde; Agilent Technologies, Santa Clara, CA, USA). The next day, samples were rinsed for 60 min in PBS and transferred to 30% sucrose/azide cryoprotectant solution overnight. Following cryoprotection, samples were orientated in cryomoulds and frozen in OCT (Agilent Technologies) using liquid nitrogen vapour and stored at −80°C. Samples were cryosectioned at 20 $\mu$m (pancreas and footpad) or 40 $\mu$m (eye and tibialis), collected on flex immunohistochemistry slides (Agilent Technologies) and allowed to air-dry for a minimum of 1 h. Sections were rehydrated in PBS for 15 min and blocked for 60 min at room temperature with 4% goat serum (Sigma-Aldrich) in PBS containing 0.4% Triton-X-100 (Sigma-Aldrich). Longitudinally sectioned tibialis muscles were incubated with fluorescent-conjugated $\alpha$-bungarotoxin[488] ($\alpha$-BTx, dilution 1:500; B13422; ThermoFisher Scientific, Scoresby, VIC, Australia) for 1 h at room temperature and post-fixed with ice-cold methanol for 30 s. Sections were incubated at 4°C in blocking solution with primary antibodies: mouse anti-$\beta$-III-tubulin (dilution 1:1000; G7121; Promega, Alexandria, NSW, Australia), mouse anti-Brn3a (dilution 1:200; MAB1585; Merck, Darmstadt, Germany), rabbit anti-glial-fibrillary acidic protein (GFAP; dilution 1:2000; Z0334; Agilent Technologies), rabbit pan-insulin (dilution 1:1000; ab8304; Abcam, Cambridge, UK), mouse anti-pan neurofilaments (SMI312; 1:400; 837904; Biolegend, San Diego, CA, USA), mouse anti-nonphosphorylated neuro-filaments (SMI32; dilution 1:400; 801701; Biolegend) and rabbit anti-synaptophysin (dilution 1:200; Ab9272; Merck). The next day, sections were thoroughly washed in PBS and incubated for 2 h at room temperature with Alexa-Fluor conjugated secondary antibodies goat anti-mouse 488 or 647 (dilution 1:1000; A11029 or A21236; Life Technologies Australia, Melbourne, VIC, Australia), goat anti-rabbit 594 (dilution 1:1000; A11037; Life Technologies Australia) in blocking solution containing 4′,6-diamidino-2-phenylindole (DAPI; 20 $\mu$g mL$^{-1}$, Sigma-Aldrich) to identify cell nuclei. Sections were thoroughly washed and coverslips mounted using fluorescence mounting medium (Agilent Technologies) and allowed to completely dry prior to imaging. For retinal, footpad and pancreas analysis, images were captured using an Olympus VSI slide scanner (Olympus, Tokyo, Japan). Next, z-stacks (1 $\mu$m) were generated using the 20× objective and converted to maximum intensity projections. For tibialis muscle analysis, z-stacks (1 $\mu$m) were generated using an Andor confocal microscope (Oxford Instruments, Oxford, UK) and NIS, version 5.02.00 (Nikon, Tokyo, Japan) and converted to maximum intensity projections for analysis.

### Analysis of pancreatic islets

Pan-insulin (insulin and pro-insulin) staining was calculated as a percentage of pancreatic islet of Langerhans area that was positive for pan-insulin immunoreactivity. Pancreatic islets were identified using DAPI staining, and pan-insulin immunoreactivity was calculated using the area of Renyi Entropy thresholded images using ImageJ, version 1.52i (NIH, Bethesda, MD, USA).

### Analysis of epidermal nerve fibre density

Epidermal nerve fibres in hindleg plantar skin were revealed using $\beta$-III-tubulin and DAPI staining to identify the border of the epidermis and dermis. Epidermal nerve fibres were identified as those that crossed into the epidermis (Lauria *et al.* 2005). The epidermal nerve fibre density was calculated by counting the number of nerve fibres that crossed the dermal–epidermal boundary and correcting for the length of the epidermis (millimetres) assessed. Epidermal fire length was measured using ImageJ (NIH). Regions of the footpad corresponding to specialised sensory structures (e.g. pads and corpuscles) were avoided when assessing epidermal nerve fibre density and length.

### Analysis of tibialis muscle neuromuscular junctions

Neuromuscular junctions in the tibialis anterior muscle were identified using $\alpha$-bungarotoxin to stain the post-synaptic ACh receptors and a combination of pan- and non-phosphorylated neurofilament antibodies and synaptophysin was used to identify nerve terminals (see above). The degree of neuromuscular junction innervation was calculated as the percentage of $\alpha$-bungarotoxin positive motor endplates that were colocalised with pre-synaptic staining.

### Analysis of retinal pathology

The different layers of the retina were identified using DAPI staining as reported previously (Daniel *et al.* 2021). The density of retinal ganglion cells (RGCs) was determined by counting the number of Brn3a positive cells in the RGC layer of the retina and expressed as cells per millimetre$^2$. The degree of GFAP infiltration into the outer layers of the retina was established by measuring the intensity of GFAP immunoreactivity within the inner plexiform layer and inner nuclear layer regions of the retina. Retinal thickness was determined by measuring from the inner vitreal surface to the inner/outer photo-receptor segment using ImageJ.

### Statistical analysis

Statistical analysis of the data was carried out using Prism, version 6.07 (GraphPad Software Inc., San Diego, CA, USA). Where data were normally distributed, as determined by the Shapiro–Wilk normality test, one-way ANOVA (single time point comparisons between groups) or two-way ANOVA (multiple comparisons between groups across time) with Student–Newman–Keuls *post hoc* test was utilised. $P < 0.05$ was considered statistically significant. Where possible, researchers were blinded to samples being assessed. Unless otherwise stated, all analysis was performed from a minimum of nine animals per experimental group. Data expressed as the mean $\pm$ SD.

## Results

### The combination of HFD and osmotic mini pump-infused STZ induces moderate hyperglycaemia with attendant increases in body weight, fat mass and fasting NEFA concentrations

Compared to control rats, all HFD fed groups had an increased body weight over the first 3 weeks of the dietary intervention ($P < 0.0001$ for all comparisons) (Fig. 2*A*). Following osmotic mini pump implants, the HFD + sham and HFD + lower STZ groups continued to gain weight at a faster rate than control rats. At the end of the 21 week intervention, the rats in the control group gained a total of $86 \pm 13$ g in body weight, whereas the HFD + sham group gained $156 \pm 26$ g ($P < 0.0001$ *vs.* control). Similar to the HFD + sham group, the rats in the HFD + lower STZ group gained $160 \pm 25$ g ($P < 0.0001$ *vs.* control; $P = 0.632$ *vs.* HFD + sham) over the course of the intervention. By contrast, the HFD + higher STZ group gained $64 \pm 62$ g ($P = 0.1005$ *vs.* control; $P < 0.0001$ *vs.* HFD + sham).

Epidydimal fat pads were excised and weighed at the end of the experiment to approximate changes in visceral adiposity (Fig. 2*C*). Compared to the control group ($11.3 \pm 1.4$ g), the HFD + sham ($23.8 \pm 3.8$ g) and HFD + lower STZ ($21.7 \pm 3.1$ g) groups had a 2-fold increase in visceral fat mass ($P < 0.0001$ for both *vs.* control). Although epidydimal fat mass was reduced in the HFD + higher STZ group ($17.3 \pm 5.9$ g) compared to HFD + sham ($P < 0.0001$), fat mass in this group remained higher than the control group ($P = 0.0020$). In accordance with the fat mass, each group of animals fed the HFD exhibited a 2-fold in the fasting NEFA concentrations at the end of the intervention (Fig. 2*D*).

Non-fasting blood glucose concentrations were assessed once per week for the 20-week intervention (Fig. 2*B*). Non-fasting glucose was similar in all groups across the first 4 weeks of the intervention. Regardless of dose, STZ increased blood glucose in HFD-fed rats ~4 days after mini pump implant ($P < 0.0001$ for both STZ groups *vs.* control; data not shown). At the completion of STZ infusion in week 6, blood glucose in the HFD + lower STZ group was ~1.5 fold higher compared to the HFD + sham group ($5.9 \pm 0.4$ *vs.* $9.1 \pm 1.6$ mmol L$^{-1}$; $P = 0.0062$). Compared to the HFD + sham group, the non-fasting blood glucose in the HFD + higher STZ group was ~2 fold higher ($5.9 \pm 0.4$ *vs.* $12.6 \pm 3.0$ mmol L$^{-1}$; $P < 0.0001$). After the surgical removal of mini pumps containing STZ, non-fasting blood glucose concentrations slowly decreased in the

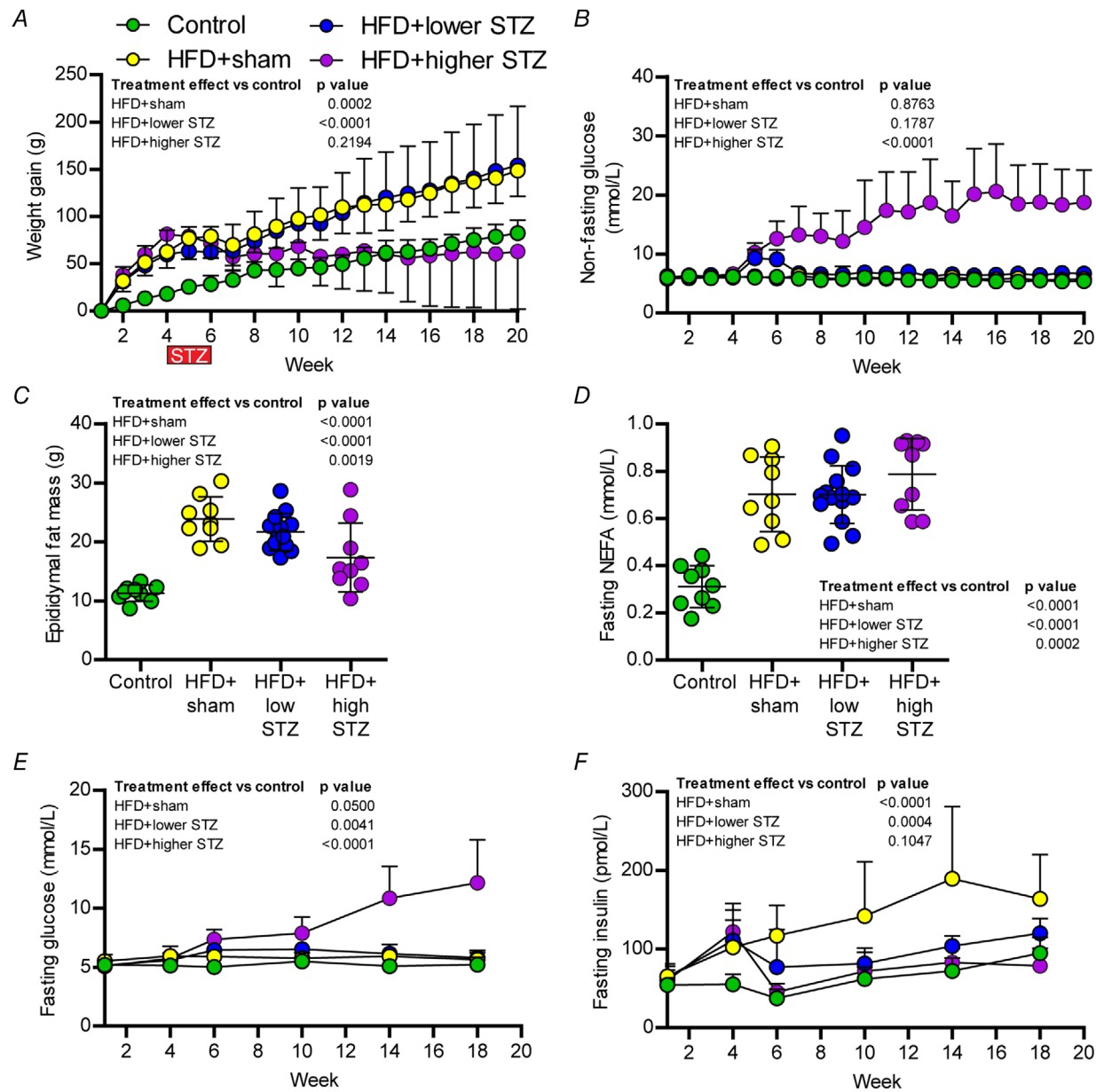

**Figure 2. Combining a HFD with osmotic mini pump-infused STZ induces long-term, dose-dependent effects on blood glucose and insulin concentrations at the same time as maintaining an obese phenotype**

*A*, average body weight gain. *B*, on-fasting blood glucose concentration across the 20-week time course. *C*, epididymal fat pat weight. *D*, fasting plasma non esterified fatty acids (NEFA) concentration at week 21. Fasting glucose and insulin concentrations were assessed prior to GTTs for weeks 1, 4, 6, 10, 14 and 18. Changes in fasting glucose and insulin across time between groups are shown in (*E*) and (*F*); data in all panels are the mean ± SD for $n = 9$ in control; $n = 9$ in HFD + sham; $n = 13$ in HFD + lower STZ; and $n = 9$ in HFD + higher STZ. Data in (*C*) and (*D*) are end-point analysis represented as the mean ± SD; each point represents data from an individual animal. In (*C*) and (*D*), one-way ANOVA with Student–Newman–Keuls *post hoc* was used to assess differences between groups. In (*A*), (*B*), (*E*) and (*F*), two-way repeated measures ANOVA with Student–Newman–Keuls *post hoc* was used to analyse data. *P* values are main effects of treatment *vs.* the control group. Within and between group interactions are outlined in the Results section. [Colour figure can be viewed at wileyonlinelibrary.com]

HFD + lower STZ group over the course of the intervention. By week 20, non-fasting blood glucose in the HFD + lower STZ group was similar to both control ($5.4 \pm 0.4$ *vs.* $6.7 \pm 0.9$ mmol $L^{-1}$; $P = 0.2776$) and HFD + sham group ($5.8 \pm 0.5$ *vs.* $6.7 \pm 0.9$ mmol $L^{-1}$; $P = 0.4363$). By contrast, non-fasting blood glucose in the HFD + higher STZ group increased slowly from week 6 to week 20 ($12.6 \pm 1.0$ *vs.* $20.0 \pm 2.3$ mmol $L^{-1}$; $P < 0.0001$) and, at the end of the intervention, was ~3-fold higher than all other groups ($P < 0.0001$ for all comparisons). Only one animal in the HFD + higher STZ group failed to follow this trend and non-fasting blood glucose in this rat decreased from 11.7 mmol $L^{-1}$ in week 6 to 8.5 mmol $L^{-1}$ in week 20.

### The combination of HFD and osmotic mini pump-infused STZ has a sustained effect on fasting blood glucose and insulin concentrations

In weeks 1, 4, 6, 10, 14 and 18, all rats underwent an overnight fast in preparation for a glucose tolerance test (GTT). Immediately prior to each GTT, blood was collected in the fasting state to assess changes in fasting glucose and insulin concentrations across the intervention (Fig. 2*E* and *F*). In the control group, fasting glucose (week 1: $5.2 \pm 0.9$ *vs.* week 18: $5.2 \pm 0.6$ mmol $L^{-1}$; $P = 0.9208$) were similar across course of the experiment, whereas fasting insulin concentrations increased (week 1: $55 \pm 10$ *vs.* week 18: $94 \pm 21$ mmol $L^{-1}$; $P = 0.0069$). There was no change in fasting glucose in the HFD + sham group over the course of the experiment ($5.5 \pm 0.5$ *vs.* $5.6 \pm 0.6$ mmol $L^{-1}$; $P = 0.7656$). By contrast, fasting insulin increased from week 4 and remained higher across the experiment relative to week 1 (week 1: $65 \pm 16$ *vs.* week 18: $164 \pm 53$ mmol $L^{-1}$; $P < 0.0001$) and the control group at week 18 ($P < 0.0001$).

Fasting glucose in the HFD + lower STZ group increased at week 6 (immediately post-STZ) relative to week 1 ($5.2 \pm 0.5$ *vs.* $6.5 \pm 0.5$ mmol $L^{-1}$; $P < 0.0001$) and remained higher for the remainder of the experiment. Fasting insulin in the HFD + lower STZ group increased from week 1 to week 4 ($64 \pm 10$ *vs.* $111 \pm 40$ pmol $L^{-1}$; $P = 0.0002$) but decreased to week 1 levels in week 6 after STZ infusion ($64 \pm 10$ *vs.* $76 \pm 17$ pmol $L^{-1}$; $P = 0.2994$). Fasting insulin in the HFD + lower STZ group increased over the course of the experiment and, by week 18, was higher than week 1 levels within the group ($64 \pm 10$ *vs.* $120 \pm 18$ pmol $L^{-1}$; $P < 0.0001$). However, fasting insulin in the HFD + lower STZ group was lower than the HFD + sham group in week 18 ($P = 0.0032$).

The most pronounced effect on fasting glucose and insulin concentrations occurred in the HFD + higher STZ group. Compared to week 1, fasting glucose increased at week 6 ($5.1 \pm 0.6$ *vs.* $7.4 \pm 0.8$ mmol $L^{-1}$; $P < 0.0001$).

There was a steady increase in fasting glucose at each subsequent time point and, by week 18, fasting glucose levels were markedly higher in this group compared to both week 1 within the group ($5.1 \pm 0.6$ *vs.* $12.2 \pm 3.6$ mmol $L^{-1}$; $P < 0.0001$) and all other groups at week 18 ($P < 0.0001$ *vs.* all other groups). Similar to all other groups fed the HFD, the HFD + higher STZ group exhibited an increase in fasting insulin at week 4 compared to week 1 ($60 \pm 18$ *vs.* $122 \pm 36$ pmol $L^{-1}$; $P < 0.0001$). After 2 weeks of higher STZ, fasting insulin decreased to week 1 levels ($60 \pm 18$ *vs.* $45 \pm 11$ pmol $L^{-1}$; $P = 0.3206$) and remained at similar levels for the remainder of the experiment, despite the animals exhibiting increasing fasting glucose concentrations across the same time period.

### The combination of HFD and osmotic mini pump-infused STZ dose-dependently reduces glucose tolerance and these effects are sustained across time

Next, we wanted to understand how the HFD and different STZ doses impacted glucose tolerance across time. As noted above, GTTs were performed on each rat at weeks 1, 4, 6, 10, 14 and 18. The complete data set for glucose changes during each GTT performed is provided in Fig. 3*A*–*D*. To compare long-term differences in glucose tolerance between groups, we have also shown blood glucose changes during the week 18 GTT for each group (Fig. 3*E*).

In the week 18 GTT, blood glucose in all groups increased by ~10 mmol $L^{-1}$ after glucose administration (Fig. 3*E*). In the control and HFD + sham groups, blood glucose peaked at 10 min ($P < 0.0001$ for both *vs.* 0 min), began declining at 20 min and returned to 0 min levels by ~105 min. Compared to the control group, the HFD + sham group exhibited a modest increase in blood glucose excursion at all time points during the GTT, although this was not significant at any time point.

Blood glucose increased in both the HFD + lower STZ and HFD + higher STZ by ~10 mmol $L^{-1}$ 10 min after glucose administration. By contrast to the control and HFD + sham groups, blood glucose continued to increase in both groups at 20 min ($P < 0.0001$ *vs.* control and HFD + sham for both). Blood glucose slowly declined in the HFD + lower STZ group from 20–120 min and by the end of the GTT remained higher than baseline levels ($6.2 \pm 0.7$ *vs.* $8.2 \pm 2.1$ mmol $L^{-1}$; $P = 0.0028$). By contrast, blood glucose in the HFD + high STZ group remained high for the first 60 min and modestly began to decline in the second hour of the GTT. At 120 min, blood glucose in the HFD + higher STZ group was higher than all other groups ($P < 0.0001$), as well as the 0 min baseline within the same group ($12.2 \pm 3.7$ *vs.* $19.9 \pm 5.4$ mmol $L^{-1}$; $P < 0.0001$)

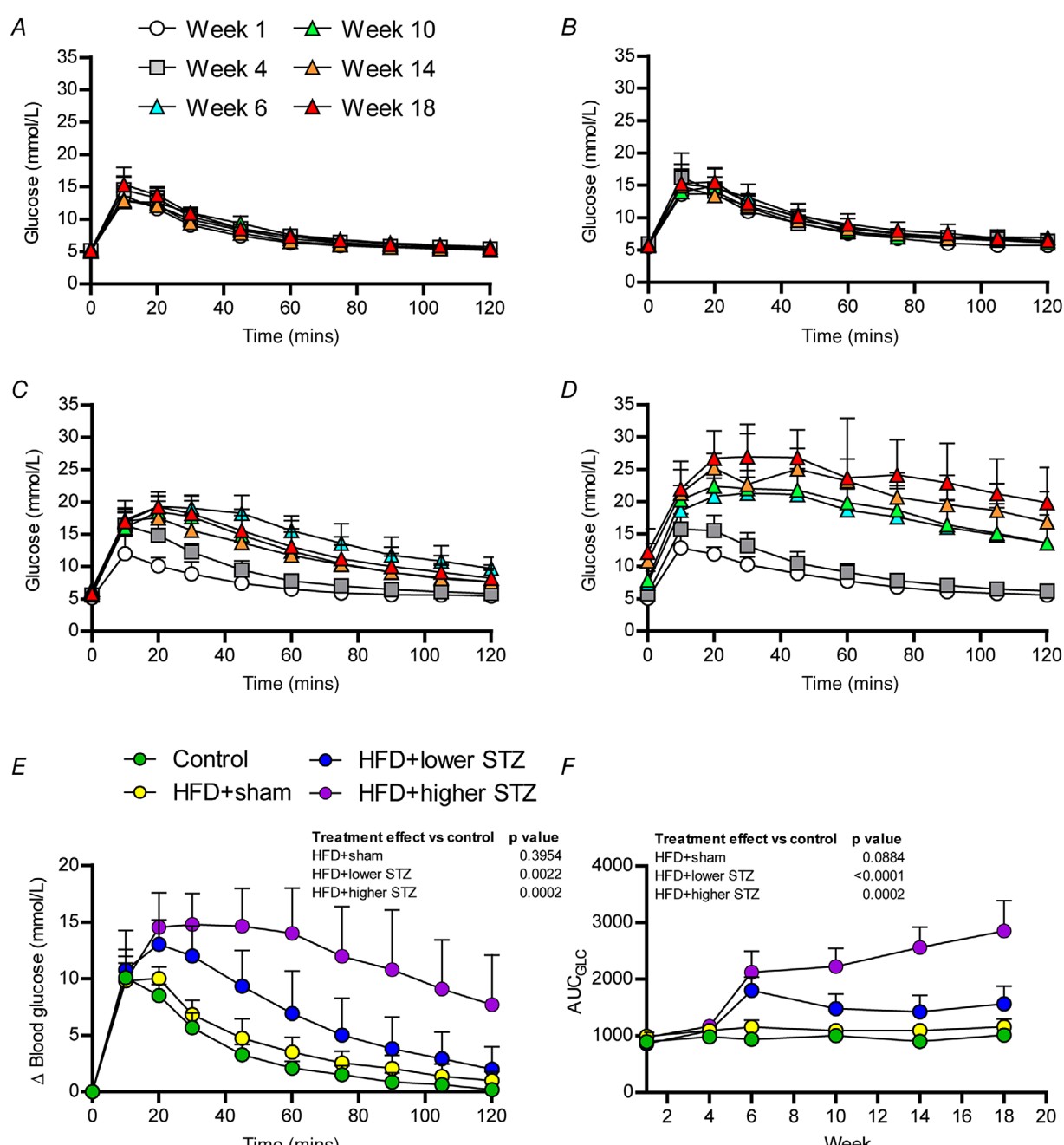

**Figure 3. Combining a HFD with and osmotic mini pump-infused STZ dose-dependently reduces glucose tolerance, and these effects are sustained over time**
Time course of blood glucose changes during each GTT across the experimental period are shown for control (*A*), HFD + sham (*B*), HFD + lower STZ (*C*) and HFD + higher STZ (*D*). *E*, change in blood glucose during the week 18 GTT for each group and represents the different groups relative glucose tolerance at the end of the study. The AUC for glucose (*F*) (AUC$_{GLC}$) was calculated from GTTs performed in weeks 0, 4, 6, 10, 14 and 18 in each group and plotted across the experimental time course. All data are shown as the mean ± SD for *n* = 9 in control; *n* = 9 in HFD + sham; *n* = 13 in HFD + lower STZ; and *n* = 9 in HFD + higher STZ. In (*E*) and (*F*), two-way repeated measures ANOVA with Student–Newman–Keuls *post hoc* was used to analyse data. *P* values are main effects of treatment *vs*. the control group. Within and between group interactions are outlined in the Results section. [Colour figure can be viewed at wileyonlinelibrary.com]

To compare the overall effects of the HFD and STZ dose on glucose tolerance between groups across the time course, we performed area under the curve (AUC)$_{Glc}$ calculations for each time point and each group and plotted this across the 20 week experimental timeline (Fig. 3F). The AUC$_{GLC}$ in the control group did not change across the experimental timeline (week 1 *vs.* week 18; $P = 0.807$). By contrast, the AUC$_{GLC}$ in the HFD + sham group increased over time and, by week, 18 was ∼20% higher than week 1 ($P = 0.0106$). In the HFD + lower STZ and HFD + higher STZ groups, the AUC$_{GLC}$ in week 6 was ∼2-fold greater than the control group ($P < 0.0001$ for both comparisons). Although the AUC$_{GLC}$ remained at this level in the HFD + lower STZ group for the remainder of the experiment, the AUC$_{GLC}$ in the HFD + higher STZ group increased across the experimental timeline and was markedly higher than all other groups in week 18 ($P < 0.0001$ for all comparisons).

### The effects of HFD and osmotic mini pump-infused STZ on glucose tolerance are mediated by dose-dependent reduction in insulin secretion and production

As well as measuring glucose during the GTT, larger volumes of blood were collected at 0, 10, 20 and 60 min during each GTT to assess plasma insulin concentration differences between the groups. The complete data set for insulin changes during each GTT is provided in Fig. 4A–D. To compare long-term differences in glucose-stimulated insulin secretion between groups, we have also shown insulin changes during the week 18 GTT for each group (Fig. 4E).

During the week 18 GTT, plasma insulin increased by 10 min ($94 \pm 21$ *vs.* $169 \pm 65$ pmol L$^{-1}$; $P = 0.0020$) in the control group and began declining toward baseline levels at 20 min ($169 \pm 65$ *vs.* $138 \pm 38$ pmol L$^{-1}$; $P = 0.1267$). By 60 min, plasma insulin in the control groups was at baseline levels ($95 \pm 21$ *vs.* $109 \pm 24$ pmol L$^{-1}$; $P = 0.4632$). Similarly, plasma insulin in the HFD + sham group increased 10 min following glucose administration ($164 \pm 53$ *vs.* $439 \pm 115$ pmol L$^{-1}$; $P < 0.0001$). Plasma insulin in the HFD + sham group remained above 0 min levels at both 20 min ($164 \pm 53$ *vs.* $308 \pm 103$ pmol L$^{-1}$; $P < 0.0001$) and 60 min ($164 \pm 53$ *vs.* $231 \pm 74$ pmol L$^{-1}$; $P = 0.0013$).

In the HFD + lower STZ group, plasma insulin increased 10 min after glucose administration ($120 \pm 18$ *vs.* $185 \pm 34$ pmol L$^{-1}$; $P < 0.0001$) but this remained ∼50% lower than the HFD + sham group at the same time ($185 \pm 34$ *vs.* $439 \pm 115$ pmol L$^{-1}$; $P < 0.0001$). Plasma insulin continued to rise in the HFD + lower STZ group at 20 min ($185 \pm 34$ *vs.* $226 \pm 36$ pmol L$^{-1}$; $P = 0.0319$) and remained at similar levels at 60 min

($226 \pm 36$ *vs.* $222 \pm 45$ pmol L$^{-1}$; $P = 0.7928$). The plasma insulin concentration at 60 min was similar between the HFD + sham and HFD + lower STZ groups ($231 \pm 74$ *vs.* $222 \pm 45$ pmol L$^{-1}$; $P = 0.6693$).

By contrast, administration of glucose in week 18 did not stimulate an increase in plasma insulin concentrations in the HFD + higher STZ group at 10 min ($79 \pm 22$ *vs.* $82 \pm 17$ pmol L$^{-1}$; $P = 0.8768$), 20 min ($79 \pm 22$ *vs.* $88 \pm 17$ pmol L$^{-1}$; $P = 0.9616$) or 60 min ($79 \pm 22$ *vs.* $85 \pm 23$ pmol L$^{-1}$; $P = 0.9471$). The lower insulin concentrations in the HFD + higher STZ group were evident despite this group having the highest glucose excursion during the GTT.

To compare the overall effects of the HFD and STZ dose on insulin secretion and production in response to a GTT, we performed AUC$_{Ins}$ calculations for each time point and each group and plotted this across time; these data are shown in Fig. 4F. All animals receiving the HFD had an increased AUC$_{Ins}$ at week 4 compared to week 1 ($P < 0.0001$ for all comparisons). Compared to the control group, the HFD + sham group had a consistently higher AUC$_{Ins}$ across the intervention ($P < 0.0001$ for all comparisons). Although the HFD + lower STZ group had higher AUC$_{Ins}$ compared to the control group at week 4 ($P < 0.0001$), in week 6, the AUC$_{Ins}$ in this group decreased and was similar to the control group in the same week ($P = 0.6117$). The AUC$_{Ins}$ in the HFD + lower STZ group increased at weeks 10, 14 and 18. At each of these time points, the AUC$_{Ins}$ in the HFD + lower STZ group was higher compared to the control group ($P < 0.0001$ for all comparisons) and lower compared to the HFD + sham group ($P < 0.0001$ for all comparisons). Lastly, the HFD + higher STZ group had an increased AUC$_{Ins}$ in week 4 compared to the control group ($P = 0.0004$). After higher STZ infusion, the AUC$_{Ins}$ in this group decreased and was similar to the control group ($P = 0.3638$) and remained at levels similar to those for the control group at weeks 10 ($P = 0.5241$), 14 ($P = 0.2633$) and 18 ($P = 0.0463$). The reduced AUC$_{Ins}$ in the HFD + higher STZ group was evident despite the marked increase in the AUC$_{Glc}$ observed in this group.

### Post-meal glucose excursions are elevated in the HFD and osmotic mini pump-infused STZ groups

During week 20, all rats underwent a meal challenge to characterise differences in post-prandial glucose excursions between the groups (Fig. 5). Blood glucose peaked at ∼45 min into the meal challenge in both the control diet ($4.8 \pm 0.3$ *vs.* $6.5 \pm 0.8$ mmol L$^{-1}$; $P = 0.0338$) and HFD + sham ($5.5 \pm 0.4$ *vs.* $7.7 \pm 0.5$ mmol L$^{-1}$; $P = 0.0088$) groups and remained relatively constant in both groups for the duration of the 120 min experiment. There were no differences in the blood

glucose concentrations between the control and HFD + sham groups at any time point. Blood glucose in the HFD + lower STZ group also peaked ~30 min into the meal challenge (6.1 ± 0.7 *vs.* 9.5 ± 2.9 mmol L$^{-1}$; $P < 0.0001$) and remained at similar levels for the rest of the experiment. Blood glucose in the HFD + higher STZ group increased to a greater extent at 30 min (16.5 ± 5.7 mmol L$^{-1}$; $P < 0.0001$ compared to all other groups) and peaked at ~60 min (11.4 ± 3.0 *vs.* 18.5 ± 5.6 mmol L$^{-1}$; $P < 0.0001$). Blood glucose in the HFD + higher STZ group remained ~18 mmol L$^{-1}$ for the rest of the 120 min experiment.

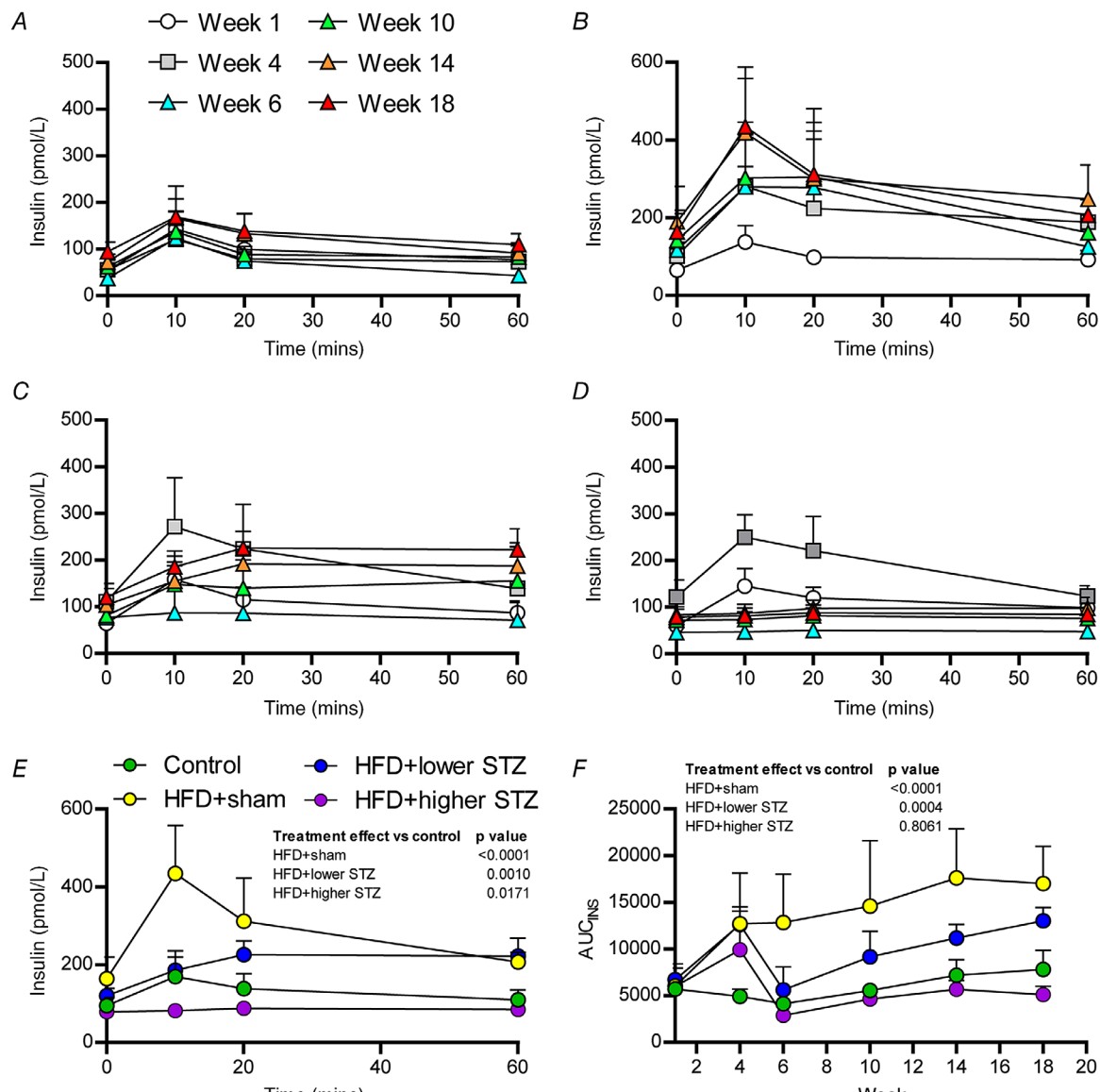

**Figure 4. Combining a HFD with and osmotic mini pump-infused STZ dose-dependently alters insulin secretion in response to a GTT**
Plasma insulin was monitored over the first 60 min of each GTT (Fig. 3) and represents the insulin secretory capacity in response to a glucose load. Time course of plasma insulin changes during each GTT across the experimental period are shown for control (*A*), HFD + sham (*B*), HFD + lower STZ (*C*) and HFD + higher STZ (*D*). The plasma insulin response between groups at week 18 GTT is shown in (*E*) and represents the different groups relative insulin secretory capacity at the end of the study. The AUC for insulin (*F*) (AUC$_{INS}$) was calculated from GTTs performed in weeks 0, 4, 6, 10, 14 and 18 in each group and plotted across the experimental time course. All data are shown as the mean ± SD for $n = 9$ in control; $n = 9$ in HFD + sham; $n = 13$ in HFD + lower STZ; and $n = 9$ in HFD + higher STZ. In (*E*) and (*F*), two-way repeated measures ANOVA with Student–Newman–Keuls *post hoc* was used to analyse data. *P* values are main effects of treatment *vs.* the control group. Within and between group interactions are outlined in the Results section. [Colour figure can be viewed at wileyonlinelibrary.com]

To compare the overall differences in post-meal glucose excursions, we performed $AUC_{GLC}$ calculations (Fig. 5*B*). There were no differences in the $AUC_{GLC}$ between the control and HFD + sham groups ($P = 0.8000$). The $AUC_{GLC}$ in the HFD + lower STZ group was ~20% higher than both control ($P = 0.0486$) and HFD + sham ($P = 0.0359$) groups. The $AUC_{GLC}$ in the HFD + higher STZ group was almost 2-fold higher than all other groups ($P = 0.0002$ *vs*. HFD + lower STZ; $P < 0.0001$ *vs*. control and HFD + sham).

## A loss of insulin secretion and production in the HFD and osmotic mini pump-infused STZ model is mediated by a dose-dependent reduction in pancreatic beta cell mass

To understand where the metabolic effects observed metabolic defects were a result of altered insulin production and secretion, pancreases were collected at the end of the study and immunohistochemistry was used to assess the islets of Langerhans (Fig. 6). The overall number of islets was calculated per mm² of pancreas (Fig. 6*B*), with no differences found between the control and HFD + sham ($P = 0.7240$), HFD + lower STZ ($P = 0.4101$) or HFD + higher STZ ($P = 0.4506$) groups. We next assessed the area of each islet and found no difference between the control group, the HFD + sham group ($P = 0.4632$), nor the HFD + lower

STZ ($P = 0.6019$) group. By contrast, we found a ~2-fold decrease in islet area in the HFD + higher STZ group compared to the control group ($24.0 \pm 10.7$ *vs*. $10.0 \pm 3.4$ $\mu$m² $P = 0.0168$) (Fig. 6*C*). In line with this, was a ~2-fold reduction in the proportion of islet area with positive immunoreactivity for pan-insulin in the HFD + higher STZ rats compared to the control ($40.4 \pm 4.5\%$ *vs*. $18.4 \pm 3.5\%$; $P = 0.0002$) (Fig. 6*D*). Compared to the control group, there was no difference in the proportion of islet area that was pan-insulin positive for in either the HFD + sham ($P = 0.1834$) or HFD + lower STZ ($P = 0.0793$) groups.

## The HFD and osmotic mini pump-infused STZ model induces peripheral neuropathy associated with a loss of epidermal nerve fibre density

To determine whether our model leads to development of peripheral neuropathy (a common complication of in those with type 2 diabetes) we performed weekly behavioural tests using the automated plantar aesthesiometer on each animal across the entire experimental protocol (Fig. 7). There was no change in the withdrawal threshold in either the control ($47 \pm 3$ *vs*. $47 \pm 4$ g; $P = 0.7321$) or HFD + sham groups ($48 \pm 3$ *vs*. $50 \pm 4$ g; $P = 0.3089$) between weeks 1 and 20. Indeed, we found no difference between the control and HFD + sham groups at any time point across the

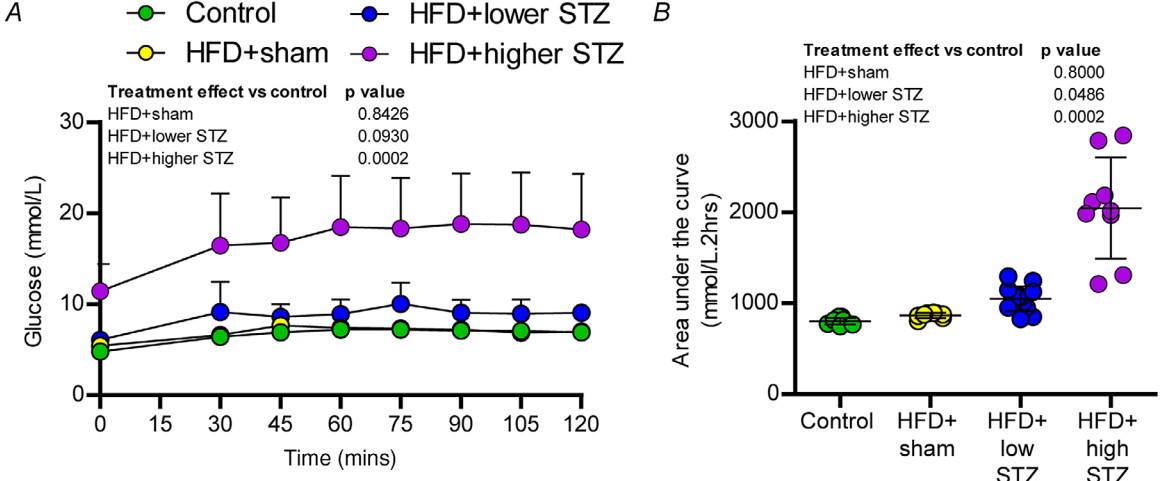

**Figure 5. Combining a HFD and osmotic mini pump-infused STZ dose-dependently increases glucose excursions following a meal challenge**
*A*, during week 20, all rats were fasted overnight and, the following morning at ~09.00 h, had their fasting glucose assessed before being provided with *ad libitum* access to their usual diet. After 30 min, the food was removed from the cage and blood glucose was monitored every 15 min for the next 90 min. *B*, AUC for glucose ($AUC_{GLC}$) during the meal challenge. All data are shown as the mean $\pm$ SD for $n = 9$ in control; $n = 9$ in HFD + sham; $n = 13$ in HFD + lower STZ; and $n = 9$ in HFD + higher STZ. In (*A*), two-way repeated measures ANOVA with Student–Newman–Keuls *post hoc* was used to analyse data. In (*B*), one-way ANOVA with Student–Newman–Keuls *post hoc* was used to assess differences between groups. *P* values are main effects of treatment *vs*. the control group. Within and between group interactions are outlined in the Results section. [Colour figure can be viewed at wileyonlinelibrary.com]

intervention. Withdrawal threshold in the HFD + lower STZ group declined steadily from week 6 and by week 9 was lower than week 1 levels (47 ± 2 *vs.* 41 ± 6 g; *P* = 0.0003). By the end of the intervention, withdrawal threshold in the HFD + lower STZ group was reduced by ~15% compared to control animals (47 ± 2 *vs.* 40 ± 4 g; *P* < 0.0001). Similarly, withdrawal threshold in the HFD + higher STZ was decreased by week 9 compared to week 1 levels (49 ± 2 *vs.* 42 ± 3 g; *P* < 0.0001). By

week 20, withdrawal threshold was ~30% lower in the HFD + higher STZ group compared to the control group (47 ± 3 *vs.* 34 ± 3 g; *P* < 0.0001) and was also lower than all other groups at the same time point (*P* < 0.0001 for all comparisons).

At the end of the intervention, we collected glabrous skin from the rear paws of these animals and used immunohistochemistry to assess changes in epidermal nerve fibre length and density (Fig. 7*B* and *C*). There were

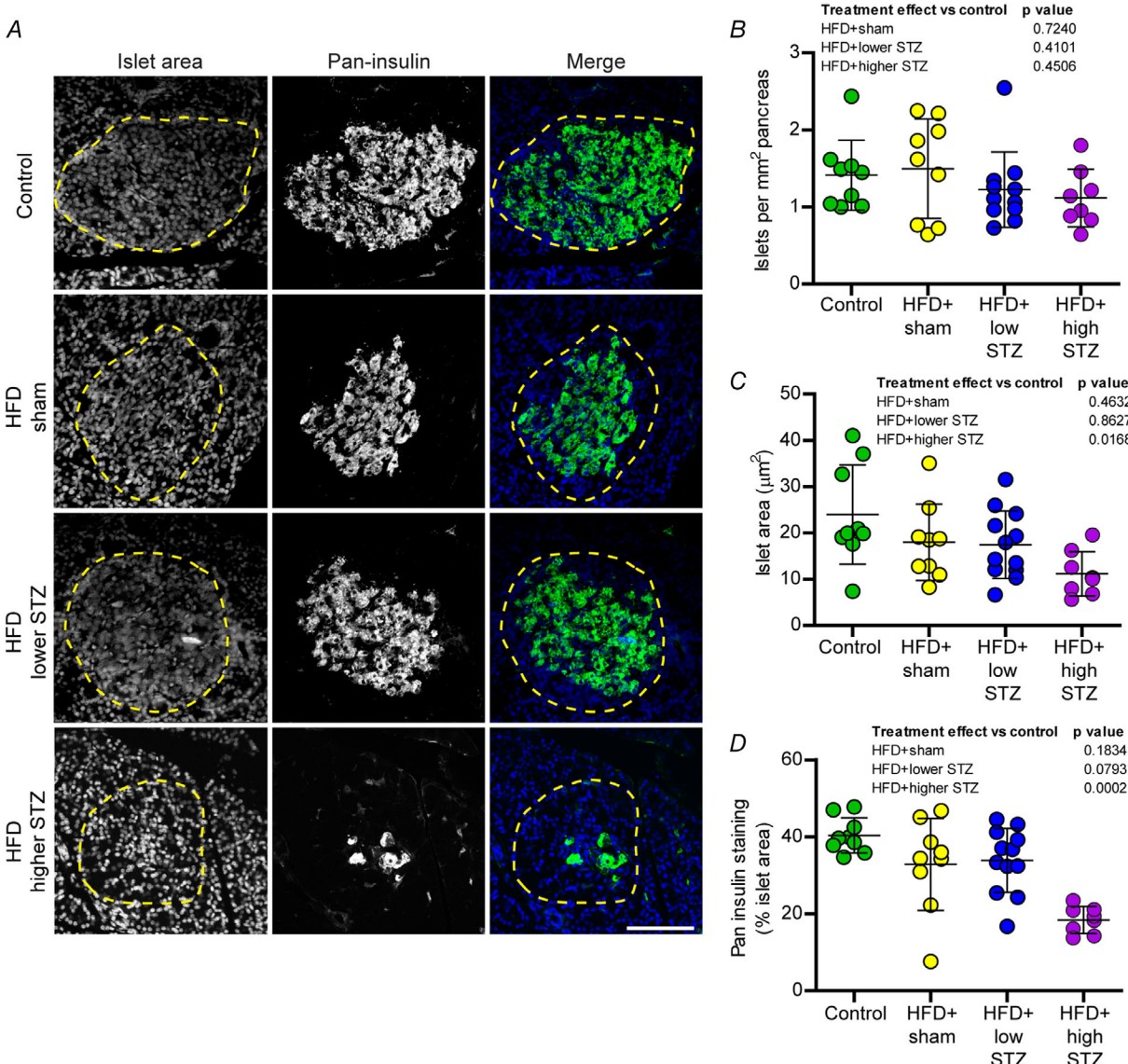

**Figure 6. Combining a HFD and osmotic mini pump-infused STZ model is mediated by a dose-dependent reduction in pancreatic beta cell mass**
*A*, pancreatic sections were immunolabelled for pan-insulin (green; labels proinsulin and insulin) and counterstained with DAPI to visualise nuclei, allowing identification of islets of Langerhans (dashed line). *B*, number of islets per mm$^2$ of pancreas across the treatment groups. *C*, average islet area. *D*, percent of islet area with positive pan-insulin immunoreactivity. Data are the mean ± SD; each point represents mean data from an individual animal. Scale bar = 50 $\mu$m. One-way ANOVA with Student–Newman–Keuls *post hoc* was used to assess differences between groups. *P* values are main effects of treatment *vs.* the control group. [Colour figure can be viewed at wileyonlinelibrary.com]

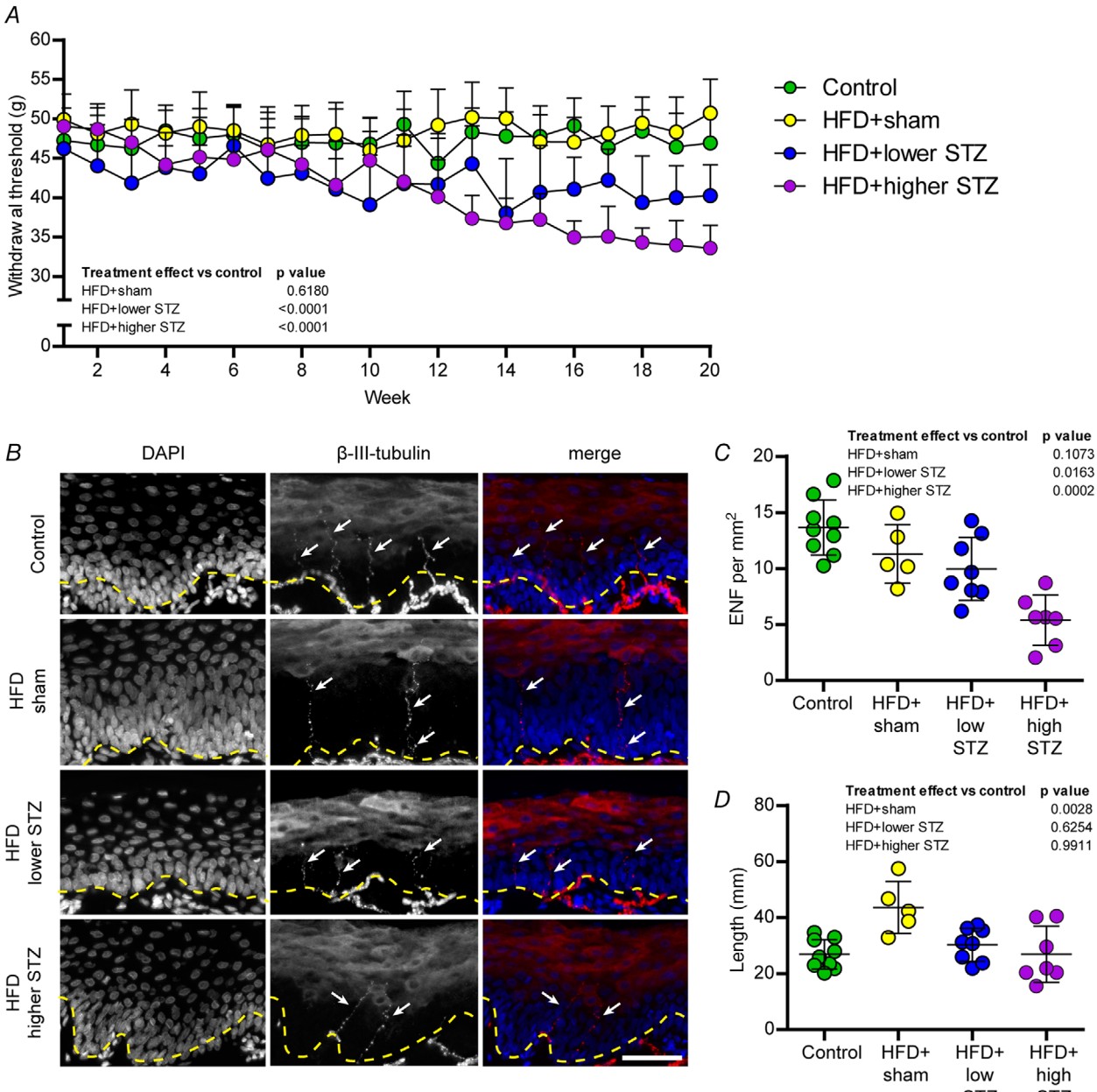

**Figure 7. The HFD and osmotic mini pump-infused STZ model dose-dependently alters peripheral hypersensitivity to touch stimulus and reduces the number of epidermal nerve fibres in rear paw glabrous skin**

*A*, pressure required to stimulate voluntary paw withdraw (withdrawal threshold) was assessed weekly using a plantar aesthesiometer. Each data point represents the average for each group and is the combined response from three separate assessments per animal per week. *B*, rear paw glabrous skin was collected in week 21 and cryosectioned for immunohistochemical analysis of epidermal nerve fibre density using *β*-III-tubulin (arrows). Counterstaining was performed using DAPI to visualise nuclei, allowing identification of the epidermal border (dashed line). Scale bar = 50 *μ*m. *C*, number of epidermal nerve fibres per mm$^2$. *D*, length of epidermal nerve fibres was assessed in each animal. In (*A*), data are the mean ± SD for *n* = 9 in control; *n* = 9 in HFD + sham; *n* = 13 in HFD + lower STZ; and *n* = 9 in HFD + higher STZ. In (*A*), two-way repeated measures ANOVA with Student–Newman–Keuls *post hoc* was used to analyse data. Individual points in (*C*) and (*D*) represent individual animals. In (*C*) and (*D*), one-way ANOVA with Student–Newman–Keuls *post hoc* was used to assess differences between groups. *P* values are main effects of treatment *vs.* the control group. Within and between group interactions are outlined in the Results section. [Colour figure can be viewed at wileyonlinelibrary.com]

no differences in epidermal nerve fibre density between the control and HFD + sham groups (13.7 ± 2.5 *vs.* 11.5 ± 1.5 fibres mm$^{-1}$; $P = 0.1073$). By contrast, the number of epidermal nerve fibres in the HFD + lower STZ group was decreased compared to the control group (13.7 ± 2.5 *vs.* 10.0 ± 2.8 fibres mm$^{-1}$; $P = 0.0163$). In line with this and the behavioural data, we found a marked decrease in nerve fibre density in the HFD + higher STZ group compared to the control group (13.7 ± 2.5 *vs.* 5.4 ± 2.2 fibres mm$^{-1}$; $P = 0.0002$). There were no differences in the length of epidermal nerve fibres in either HFD + lower STZ ($P = 0.6254$) or HFD + higher STZ ($P = 0.9911$) rats compared to the control group. However, we observed an increase in epidermal nerve fibre length in HFD + sham group compared to the control group (26.9 ± 5.3 *vs.* 43.6 ± 9.3 $\mu$m; $P = 0.0028$).

### The HFD and osmotic mini pump-infused STZ model also exhibits reduced motor neuron innervation of the tibialis anterior muscle

To investigate whether the peripheral neuropathy in the present model was specific to the sensory axons of the footpad or whether the neuropathy affected other peripheral neurons, analysis of neuromuscular junction innervation was performed in the tibialis anterior muscle. Innervation of the neuromuscular junction was defined as colocalization between the pre-synaptic nerve terminal components (neurofilaments and synaptophysin) and the post-synaptic ACh receptors (Fig. 8). Compared to the control group, we found no difference in proportion of neuromuscular junctions that were occupied by pre-synaptic components in HFD + sham group (91.2 ± 9.8 *vs.* 82.5 ± 10.2% innervation; $P = 0.2956$). Surprisingly, we demonstrated an overall reduction in the proportion of neuromuscular junctions that were occupied by pre-synaptic components in HFD + lower STZ rats compared to the control group (91.2 ± 9.8 *vs.* 69.5 ± 18.9% innervation; $P = 0.0120$). In line with this, we identified a similar reduction in neuromuscular junction innervation in HFD + higher STZ rats compared to the control group (91.2 ± 9.8 *vs.* 66.4 ± 19.8% innervation; $P = 0.0148$). Interestingly, these differences were apparent without a change in the total number of neuromuscular junctions between any groups (data not shown).

### The HFD and osmotic mini pump-infused STZ model does not affect the density of retinal ganglion neurons, nor the thickness of the retina

In the present study, we attempted to quantify vision loss across in animals exposed to HFD + lower and HFD + higher STZ using the optokinetic reflex as

reported previously (Daniel *et al.* 2021). However, because Sprague–Dawley rats are an albino strain, they inherently have poor visual acuity. Thus, we were unable to successfully perform this behavioural assessment in the present study. Nevertheless, to assess whether the neurological effects of our model extended to the retina, the eyes were collected and the retinal thickness and density of RGCs was calculated at the end of the intervention (Fig. 9). There was no difference in the thickness of the retina between the control and HFD + sham ($P = 0.9668$), HFD + lower STZ ($P = 0.9630$) and HFD + higher STZ ($P = 0.9709$) groups. Similarly, there was no difference

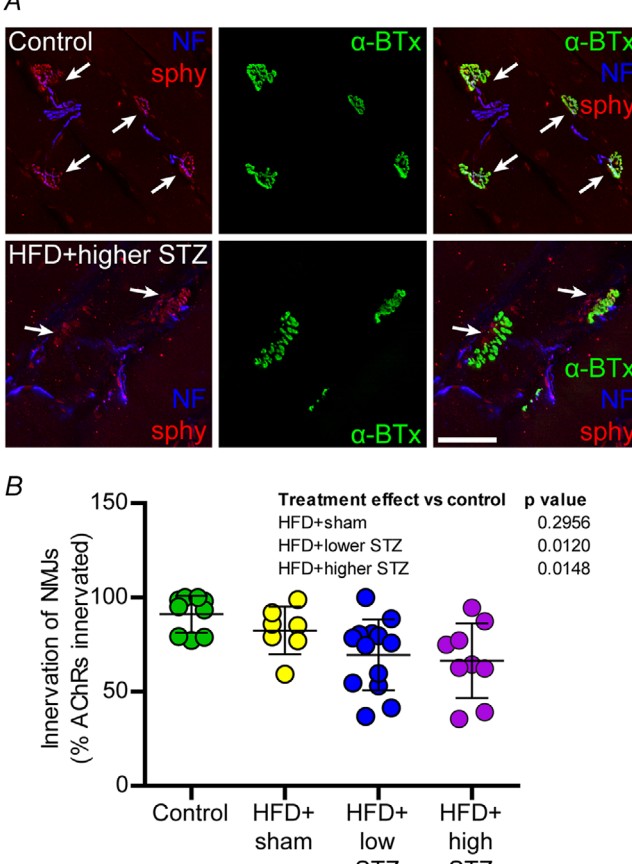

**Figure 8. The HFD and osmotic mini pump-infused STZ model exhibits loss of normal motor endplate innervation in the tibialis anterior muscle**

*A*, tibialis muscle sections were labelled for $\alpha$-bungarotoxin ($\alpha$-BTx, green) to label the post-synaptic ACh receptors and phosphorylated and non-phosphorylated neurofilaments (blue) and synaptophysin (sphy, red) to identify the pre-synaptic nerve terminal. Arrows indicate colocalised pre- and post-synaptic elements. *B*, percent of AChRs innervated was assessed from at least two muscle sections from each animal. Data are the mean ± SD; each point represents mean data from an individual animal. Scale bar = 50 $\mu$m. One-way ANOVA with Student–Newman–Keuls *post hoc* was used to assess differences between groups. *P* values are main effects of treatment *vs.* the control group. [Colour figure can be viewed at wileyonlinelibrary.com]

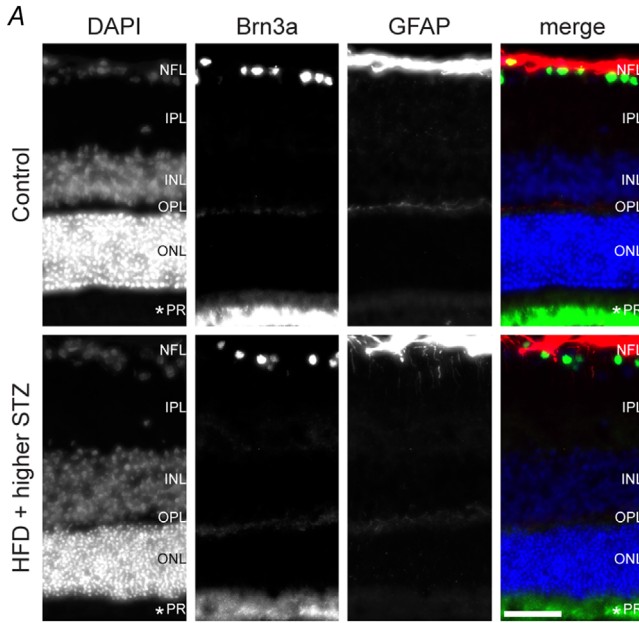

IPL, inner plexiform layer; INL, inner nerve layer; OPL, outer plexiform layer; ONL, outer nerve layer; PR, photoreceptor layer. An asterisk (∗) indicates inner/outer photoreceptor segment. *B*, total retinal thickness. *C*, number of RGCs per mm². Data are the mean ± SD; each point represents mean data from an individual animal. Scale bar = 50 μm. One-way ANOVA with Student–Newman–Keuls *post hoc* was used to assess differences between groups. *P* values are main effects of treatment *vs.* the control group. [Colour figure can be viewed at wileyonlinelibrary.com]

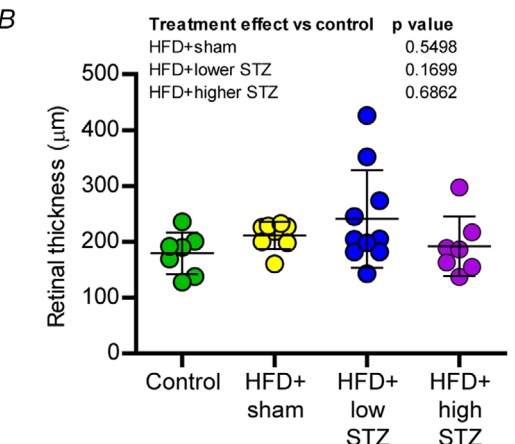

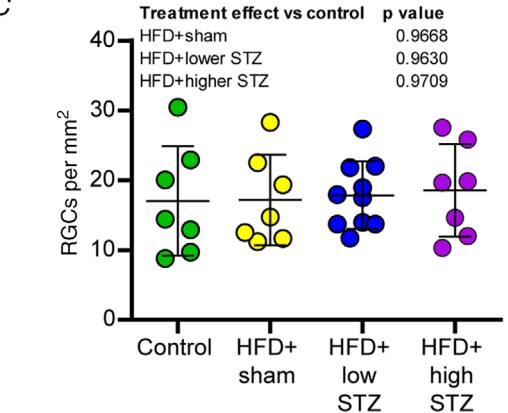

**Figure 9. The HFD and osmotic mini pump-infused STZ model does not exhibit signs of retinal degeneration**
*A*, retinas were cross-sectioned and immunolabelled for retinal ganglion cells (RGCs, Brn3a, green), astrocytes (GFAP, red) and counterstained with DAPI to visualise nuclei and enable identification of retinal layers. NFL, nerve fibre and retinal ganglion cell (RGC) layer;

in the number of RGCs per mm² of the RGC retinal layer between the control and HFD + sham ($P = 0.5498$), HFD + lower STZ ($P = 0.1699$) and HFD + higher STZ ($P = 0.6862$) groups. There was no change in the distribution of GFAP immunoreactivity between the treatment groups (data not shown).

## Discussion

A major impediment in the field of type 2 diabetes research has been the lack of appropriate animal models that reproduce human-like type 2 diabetes. In the present study, we have developed and comprehensively phenotyped a rodent model of type 2 diabetes that is able to recapitulate distinct stages of type 2 diabetes characterised by moderate hyperglycaemia, hyperinsulinaemia, impaired glucose tolerance, dyslipidaemia and obesity, which comprise the cluster of metabolic abnormalities that underpin type 2 diabetes (Talbot *et al.* 2012; Kleinridders *et al.* 2014; Biessels & Reagan, 2015; Verdile *et al.* 2015). We have also confirmed that these animals exhibit signs of progressive peripheral neurodegeneration, a disorder commonly seen in humans with type 2 diabetes. Therefore, this new model provides an opportunity to improve our understanding of the mechanistic links between type 2 diabetes and complications such as neurodegeneration, as well as interrogation of novel drugs that target disease-causing mechanisms.

Obesity is a characteristic and primary risk factor for development of type 2 diabetes. This has largely been attributed to development of impaired glucose tolerance that drives insulin resistance in organs such as skeletal muscles and the liver (Kahn *et al.* 2006; DeFronzo, 2007; Premilovac *et al.* 2013). In the initial stages, and as insulin resistance continues, the body can compensate and maintain normoglycaemia by producing more insulin. However, without intervention and continuing insulin resistance, insulin production is limited by beta cell function and hyperglycaemia develops. This is the classic pathophysiological progression of human type 2 diabetes (DeFronzo, 2007; DeFronzo *et al.* 2015) and this decade long preclinical time course cannot be exactly modelled in animals. For this reason, the HFD + STZ model has traditionally been utilised to induce obesity (HFD) and

beta cell dysfunction (STZ) to accelerate development of hyperglycaemia (Srinivasan *et al.* 2005; Skovsø, 2014). Recently, we built upon this model by demonstrating that the combination of HFD + osmotic mini pump-infused STZ enables targeted induction of hyperglycaemia and glucose intolerance in the short-term (Premilovac *et al.* 2017). In the present study, we show that these metabolic effects persist long-term and that this methodology can be used to induce distinct stages of type 2 diabetes in rats, dependent on the dose of STZ utilised.

Combining the HFD with a lower dose of STZ (100 mg kg$^{-1}$), we demonstrate sustained induction of obesity, hyperglycaemia, hyperlipidaemia and modest hyperinsulinaemia. These are characteristic features of early-stage human type 2 diabetes where insulin production is increased in a compensatory manner but there is insufficient insulin to effectively regulate blood glucose concentrations, leading to hyperglycaemia, particularly in the post-prandial state (Kahn *et al.* 2006; DeFronzo *et al.* 2015; Russell *et al.* 2017). In accordance, we show that these animals have impaired glucose tolerance and increased post-prandial glucose excursions at the same time as maintaining a relatively high insulin secretory capacity across the experimental timeline. The ability to induce glucose intolerance, maintain obesity and maintain a relatively high insulin secretory capacity is a major advance in the field given that other HFD + STZ models typically lead to hyperglycaemia as a result of an almost complete loss of insulin secretory capacity (Srinivasan *et al.* 2005; Zhang *et al.* 2009). This is a key distinguishing feature of the current model in which we can reproduce the complex metabolic disturbances associated with early-stage type 2 diabetes in humans where both cardiovascular and neurological complications are proposed to begin (Brownlee, 2001; DeFronzo, 2007; Feldman *et al.* 2019).

Combining the HFD with a higher dose of STZ (120 mg kg$^{-1}$) resulted in the development of frank hyperglycaemia and hypoinsulinaemia (but not loss of insulin) at the same time as the retention of an obese and hyperlipidaemic phenotype. Thus, the HFD + higher STZ model is akin to later stage human type 2 diabetes where the beta cells produce some, but not sufficient, insulin to effectively regulate glucose homeostasis (DeFronzo *et al.* 2015). In response to a GTT or meal challenge, glucose levels in these animals stay elevated beyond 2 h, whereas insulin production remains low, indicating a substantial loss of pancreatic beta cells. We confirmed this through histological assessment of islets of Langerhans where this group had a marked reduction in both total islet area as well as insulin staining within islets. Thus, the higher STZ dose has a more pronounced effect on the pancreas and, as a result, this led to more overt metabolic defects as seen in people with established type 2 diabetes where drug interventions are normally prescribed (DeFronzo *et al.* 2015;

Russell *et al.* 2017). It should be noted that this higher dose of STZ induces beta cell dysfunction by week 6, which progressively gets worse across the 20 week intervention. Nevertheless, the rats exposed to this dose retain some insulin producing capability and this is in contrast to STZ injection models where animals transition from normal insulin production to complete insulin deficiency in a matter of hours (Srinivasan *et al.* 2005). Thus, our approach is a significant advance also for modelling later stages of type 2 diabetes (i.e. during the phase of declining beta cell function) that ultimately provides a powerful platform for testing of new therapeutics targeting beta cell function, as well as late-stage diabetic complications.

Next, we wanted to understand whether these animals developed the classic neurological complications of type 2 diabetes such as peripheral neuropathy, which is a major cause of morbidity that effects as many as 50% of people with type 2 diabetes across the course of their disease (Ziegler *et al.* 1992; Dyck *et al.* 1993; Won *et al.* 2014). Diabetic peripheral neuropathy is reported to affect a wide range of neuron types, including sensory, motor and autonomic fibres (Sinnreich *et al.* 2005; Feldman *et al.* 2019). Sensory neuropathy primarily affects the small sensory fibres of the hands and feet in a stocking/glove distribution (Palumbo *et al.* 1978; Sinnreich *et al.* 2005; Feldman *et al.* 2019) and typically manifests as neuropathic pain, hyperalgesia, allodynia and numbness (Dyck *et al.* 1993; Feldman *et al.* 2019). Importantly, both the HFD + lower STZ and HFD + higher STZ groups exhibited increased sensitivity to touch/pressure stimulus, a behavioural sign of diabetic peripheral neuropathy (Lauria *et al.* 2005; Coppey *et al.* 2018). This was confirmed through assessment of hind paw withdrawal threshold where both the HFD + lower STZ and HFD + higher STZ groups demonstrated reduced withdrawal threshold by week 9. Importantly, withdrawal threshold progressively declined in the animals that exhibited progressive hyperglycaemia, suggesting a further loss of peripheral nerve function in these animals. We confirmed the presence of epidermal nerve degeneration through histological assessment of the rear paw glabrous skin and show a dose-dependent decline in epidermal nerve fibre density. A number of publications have demonstrated development of peripheral neuropathy in the HFD + STZ model where reduced sensitivity is noted with reduced epidermal nerve fibre density (Coppey *et al.* 2018). By contrast to these reports, we demonstrate progressive peripheral touch hypersensitivity and a reduction in epidermal nerve fibre density. The discrepancy between our results and previous reports may reflect that our model is a milder version of type 2 diabetes where painful neuropathies are typically noted in human patients (Vincent *et al.* 2011; Callaghan *et al.* 2012; Pop-Busui *et al.* 2017). Importantly, although all of the type 2 diabetic animals in our study exhibited some level of hypersensitive peripheral neuropathy, it should also be

noted that not all individuals with type 2 diabetes develop peripheral neuropathy. Although we do not fully understand the mechanisms that cause painful/hypersensitive or loss of sensation neuropathies in type 2 diabetes, our model provides an excellent platform for comprehensively characterising these important phenotypes.

To understand whether other components of the peripheral nervous system are also affected in our model, we analysed motor neuron innervation of the lower leg muscles in our animals. Interestingly, our data demonstrate a reduction in the extent of endplate innervation by motor neurons in the tibialis anterior muscle in both the HFD + lower and HFD + higher STZ groups. To our knowledge, this has not been reported previously in the literature and indicates that the neurodegeneration seen in type 2 diabetes is not confined to the sensory neurons of the skin but also extends to other components of the peripheral nervous system. Although the functional consequences of this were not assessed in the present study, a loss of motor neuron function may help explain reports of reduced muscle strength (Bianchi & Volpato, 2016) and reduced motor nerve conduction velocities in diabetic patients (Wilson & Wright, 2014; Kobori *et al.* 2017). Further work is required to understand the mechanisms that lead to loss of motor neuron innervation in muscles and the consequences of this on muscle function in type 2 diabetes.

Lastly, we wanted to assess whether our type 2 diabetic animals also develop signs of diabetic retinopathy, a leading cause of blindness worldwide (Bourne *et al.* 2013; Lee *et al.* 2015; Zafar *et al.* 2019). Unfortunately, because we utilised an albino rat strain in our experiments, we were unable to directly assess changes in visual acuity across our experiment as previously published (Daniel *et al.* 2021). Nevertheless, we collected eyes from all animals at the end of the time course and used histological techniques to assess changes in the retina as previously described. We show no change in retinal thickness or the number of RGCs in either the HFD + lower or HFD + higher STZ groups. Thus, the results from this model indicate that peripheral neurodegeneration occurs prior to changes within a more protected organ such as the retina, a notion that is broadly supported in the literature (Pop-Busui *et al.* 2017). It should be noted, however, that we did not assess other markers of retinal degeneration such as microvascular structure and function and this is a limitation of the present study.

Although the model described in the present study provides an advance in the field, the HFD + STZ model itself remains artificial regarding human type 2 diabetes pathogenesis. Although our model involves a background of HFD to induce obesity-associated insulin resistance seen in humans, the administration of STZ does not replicate the exact cause of beta cell death as seen in humans. Nevertheless, the result of both human

type 2 diabetes progression and STZ administration in our model is similar, advancing metabolic dysregulation that ultimately drives the pathological features of type 2 diabetes, as well as peripheral neurodegeneration. Another limitation of the work is the timeframe during which the type 2 diabetic animals develop signs of peripheral neuropathy. Although our animals develop signs of hypersensitive neuropathy after 9–10 weeks, in humans with type 2 diabetes peripheral neuropathy typically takes 5–10 years to clinically manifest. The difference in development and progression of peripheral neuropathy between our animal model and that observed in humans may be a result of the level and quality of glucoregulatory control. Humans are typically prescribed glucose lowering medications such as metformin, which may delay the development of neurodegeneration. By contrast, our animals are not medicated and have chronic hyperglycaemia, which may drive accelerated peripheral neurodegeneration. Therefore, although our model recapitulates the neurodegenerative complications of type 2 diabetes, the timeframe during which this occurs does not represent the human course of the disease.

In conclusion, we report the development of a new rat model where the dose of STZ delivered by osmotic mini pumps in combination with a long-term HFD can be used to model different stages of type 2 diabetes. Using the HFD + lower STZ dose, we report the induction of sustained hyperglycaemia, hyperinsulinaemia, reduced glucose tolerance, obesity and hyperlipidaemia, which are hallmarks of established but not overt type 2 diabetes seen in humans. Using the HFD + higher STZ dose, we report the induction of hyperglycaemia, impaired glucose tolerance, hyperlipidaemia and continued decline in beta cell insulin-production, which are hallmarks of established type 2 diabetes where patients are typically diagnosed. Importantly, both groups exhibited the classic signs of peripheral neurodegeneration as seen in humans with type 2 diabetes. Therefore, this model recapitulates the key metabolic disturbances seen in type 2 diabetes and provides an excellent platform for testing new therapeutics to prevent or reverse complications of type 2 diabetes such as peripheral neuropathy.

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

## Additional information

### Data availability statement

All data are reported in the published article.

### Competing interests

The authors declare that they have no competing interests.

### Author contributions

All experiments were performed at the Medical Sciences Precinct, University of Tasmania. DP, LF and BT conceived and designed the study. DP, CS and AD performed the surgeries, glucose tolerance tests, behavioural analysis and general husbandry of animals across the study. KS performed post-mortem tissue collection and histological analysis. DP and KS analysed/interpreted the data and drafted the manuscript. All authors revised the manuscript and provided intellectual feedback. All authors have read and approved the final version of this manuscript submitted for publication. All authors agree to be accountable for all aspects of the work in ensuring that questions related to the accuracy or integrity of any part of the work are appropriately investigated and resolved. All persons designated as authors qualify for authorship, and all those who qualify for authorship are listed.

### Funding

This research was funded by the National Health and Medical Research Council (NHMRC) of Australia grant APP1088952 and the University of Tasmania Research Enhancement Grant Scheme (REGS).

Open access publishing facilitated by University of Tasmania, as part of the Wiley – University of Tasmania agreement via the Council of Australian University Librarians.

### Acknowledgements

We acknowledge Mr Oleksandr Dorohokuplia and the entire Animal Services team at the University of Tasmania for assisting with the care of the animals used in the present study.

### Keywords

high fat diet, insulin resistance, obesity, peripheral neuropathy, streptozotocin, type 2 diabetes

## Supporting information

Additional supporting information can be found online in the Supporting Information section at the end of the HTML view of the article. Supporting information files available:

**Statistical Summary Document**
**Peer Review History**

