## [Peer Review History · The Journal of Physiology]

Development and characterisation of a rat model that exhibits both metabolic dysfunction and neurodegeneration seen in type 2 diabetes

Katherine Southam, Chantal de Sousa, Abraham Daniel, Bruce Taylor, Lisa Foa, and Dino Premilovac

DOI: 10.1113/JP282454

Corresponding author(s): *Dino Premilovac (dino.premilovac@utas.edu.au)*

The following individual(s) involved in review of this submission have agreed to reveal their identity: Bettina Mittendorfer (Referee #1)

Review Timeline:

Submission Date:	12-Oct-2021
Editorial Decision:	29-Nov-2021
Revision Received:	16-Dec-2021
Editorial Decision:	20-Jan-2022
Revision Received:	23-Jan-2022
Accepted:	02-Feb-2022

Senior Editor: Kim Barrett

Reviewing Editor: Maria Chondronikola

Transaction Report:

Dear Dr Premilovac,

Re: JP-TFP-2021-282454 "Development and characterisation of a rat model that exhibits both metabolic dysfunction and neurodegeneration seen in humans with type 2 diabetes" by Katherine Southam, Chantal de Sousa, Abraham Daniel, Bruce Taylor, Lisa Foa, and Dino Premilovac

Thank you for submitting your manuscript to The Journal of Physiology. It has been assessed by a Reviewing Editor and by 2 Referees and the reports are copied below.

I regret to say that the manuscript has not been accepted for publication.

Some positive comments were made on the manuscript. Unfortunately, they did not outweigh the more serious criticisms which led the Reviewing Editor to recommend rejection.

I am sorry to have to pass on this disappointing news, and hope it will not discourage you from making future submissions of new work to The Journal of Physiology.

However, we believe your manuscript is worthy of further consideration and suggest that you transfer your manuscript to Physiological Reports (<https://physoc.onlinelibrary.wiley.com/hub/journal/2051817X/aims-and-scope/read-full-aims-and-scope>), a peer-reviewed, open access, interdisciplinary journal, jointly owned by the American Physiological Society and The Physiological Society.

To transfer your manuscript to Physiological Reports, the corresponding author must send authorization within 120 days of receipt of this letter. Please use this link Transfer to Physiological Reports to send an authorization email to transfer your manuscript. If your manuscript does not require additional peer review, the editors of Physiological Reports will aim to give you an initial decision within 3 working days. In fact, >80% of transferred submissions are accepted for publication. Please note, of course, that we cannot guarantee final acceptance.

I hope you will take advantage of the opportunity to allow the editors of Physiological Reports to evaluate your manuscript.

You may be able to publish Open Access with no direct cost to yourself. You can check your eligibility here <https://secure.wiley.com/openaccess?>

Yours sincerely,

Professor Kim E. Barrett
Editor-in-Chief
The Journal of Physiology
<https://jp.msubmit.net>
<http://jp.physoc.org>
The Physiological Society
Hodgkin Huxley House
30 Farringdon Lane
London, EC1R 3AW
UK
<http://www.physoc.org>
<http://journals.physoc.org>

EDITOR COMMENTS

Reviewing Editor:

Thank you for submitting your manuscript for publication to Journal of Physiology. The manuscript is well-written and the results of study are comprehensive. Although the investigated model appears to more closely recapitulate the human condition than the traditionally used animal models, the study did not receive adequate support from the reviewers. Further, this manuscript does not provide new mechanistic insights on the pathophysiology of the obesity-related metabolic abnormalities. Accordingly, we believe that this manuscript may be more appropriate for publication to Physiological Reports.

REFeree COMMENTS

Referee #1:

The authors developed a new rat model that exhibits both metabolic dysfunction and neurodegeneration seen in humans with type 2 diabetes. This was accomplished by using a high fat/high sugar diet and delayed-onset infusion of streptozotocin to dose-dependently destroy pancreatic beta cell mass. The authors found that the combination of the diet and low dose streptozotocin leads to obesity combined with moderate fasting hyperglycaemia, hyperinsulinaemia, and impaired glucose tolerance. Combining the diet with high dose streptozotocin leads to hypoinsulinaemia (but not insulin depletion) and severe hyperglycemia (fasted and postprandially). Both models (low and high dose streptozotocin) demonstrated signs of peripheral neurodegeneration in the skin and muscle.

This model offers advantages over existing models that either lead to quick destruction of beta cell mass or fail to cause hyperglycemia and impaired glucose tolerance. It is expected that this new rat model will be useful for the study of metabolic and neurological alterations in the progression from normoglycemia to hyperglycemia-hyperinsulinemia to type 2 diabetes.

The experimental design is appropriate. The characterization of metabolic function is detailed and the addition of features of neuropathy is valuable. The paper is overall well written, although the results are a little lengthy, and the data are nicely presented.

I have no specific comments. The authors did a great job, aside from the somewhat lengthy results section.

Referee #2:

Southam et al. sought to induce different stages of type 2 diabetes (T2DM) using a high fat/high sugar diet and 14-day infusion of streptozotocin to dose-dependently destroy pancreatic beta cell mass. During a four month follow-up period, the authors attempted to characterise this model and demonstrated that they develop sustained metabolic dysfunction and progressive peripheral neurodegeneration.

The authors appear to have performed a robust set of experiments and characterisation work in this current manuscript, and should be commended for their experimental and technical skill.

The authors are correct that current models of T2DM do not fully recapitulate the disease progression, and symptoms observed in patients. However, I do feel that the author's claim that this model is able to "reproduce the complex metabolic disturbances associated with early-stage type 2 diabetes in humans" is overstated given the relatively limited characterisation work performed in this rodent model (compared to the totality of human T2DM characterisation and investigation). Additionally, attempting to recreate the full spectra of symptoms and co-morbidities observed in human disease, is for me, of questionable value given recent work demonstrating that T2D is a highly heterogenous disease (e.g. <https://doi.org/10.2337/dbi20-0001>). Thus my main comments are related to tempering of certain claims made by the authors.

Comments

Title: "Development and characterisation of a rat model that exhibits both metabolic 2 dysfunction and neurodegeneration seen in humans with type 2 diabetes" I don't think that the manner of metabolic dysfunction and neurodegeneration is precisely observed in humans. Therefore the authors should remove "seen in humans with type 2 diabetes" from the title.

"we wanted to understand whether these animals developed the classic neurological complications of type 2 diabetes such as peripheral neuropathy - a major cause of morbidity that effects approximately 50% of people with type 2 diabetes" The prevalence of neuropathy in T2DM appears to vary in the literature e.g. Feldman et al., 2019 suggests 8-51% prevalence. Perhaps the authors could briefly discuss the range of reported prevalence, and any relevance this might have to their model.

The authors appear to highlight certain aspects of their model's timeline of symptom progression when it suits, and 'play-down' other aspects when it doesn't. E.g. altered hind paw withdrawal threshold was observed already at week 9, whereas, in the human condition neuropathic symptoms are progressive and occur secondary, often over a long period of time, to poor glucoregulatory control. From Feldman et al., 2019 : "Indeed, the prevalence of diabetic neuropathy increased from 8% to 42% in patients with T2DM when patients were monitored for 10 years". The authors should be clear that the timeline of development of peripheral neuropathy in their model does not recapitulate disease progression in many people with T2DM.

Given the comments above, it is clear that this concluding sentence is overstated and should be revised: "Therefore, this model recapitulates the complex and multiple metabolic disturbances seen in people with type 2 diabetes that will improve our understanding of the mechanisms that lead to neurodegeneration as well as providing a powerful platform for testing new therapeutics to limit nervous system damage in type 2 diabetes."

Re: JP-TFP-2021-282454 "Development and characterisation of a rat model that exhibits both metabolic dysfunction and neurodegeneration seen in humans with type 2 diabetes" by Katherine Southam, Chantal de Sousa, Abraham Daniel, Bruce Taylor, Lisa Foa, and Dino Premilovac

Dear Professor Barrett,

The authors would like to thank the reviewers and editors for giving us the opportunity to revise and resubmit our manuscript to the Journal of Physiology. Below is a point-by-point summary of the alterations that have been made to the manuscript based on the review comments provided.

Reviewer 1.

The authors developed a new rat model that exhibits both metabolic dysfunction and neurodegeneration seen in humans with type 2 diabetes. This was accomplished by using a high fat/high sugar diet and delayed-onset infusion of streptozotocin to dose-dependently destroy pancreatic beta cell mass. The authors found that the combination of the diet and low dose streptozotocin leads to obesity combined with moderate fasting hyperglycaemia, hyperinsulinaemia, and impaired glucose tolerance. Combining the diet with high dose streptozotocin leads to hypoinsulinaemia (but not insulin depletion) and severe hyperglycemia (fasted and postprandially). Both models (low and high dose streptozotocin) demonstrated signs of peripheral neurodegeneration in the skin and muscle.

This model offers advantages over existing models that either lead to quick destruction of beta cell mass or fail to cause hyperglycemia and impaired glucose tolerance. It is expected that this new rat model will be useful for the study of metabolic and neurological alterations in the progression from normoglycemia to hyperglycemia-hyperinsulinemia to type 2 diabetes.

The experimental design is appropriate. The characterization of metabolic function is detailed and the addition of features of neuropathy is valuable. The paper is overall well written, although the results are a little lengthy, and the data are nicely presented.

I have no specific comments. The authors did a great job, aside from the somewhat lengthy results section.

Authors response: we thank reviewer 2 for the positive comments provided regarding our work.

Changes to manuscript: No changes have been made.

Reviewer 2.

Southam et al. sought to induce different stages of type 2 diabetes (T2DM) using a high fat/high sugar diet and 14-day infusion of streptozotocin to dose-dependently destroy pancreatic beta cell mass. During a four-month follow-up period, the authors attempted to characterise this model and demonstrated that they develop sustained metabolic dysfunction and progressive peripheral neurodegeneration.

The authors appear to have performed a robust set of experiments and characterisation work in this current manuscript, and should be commended for their experimental and technical skill.

The authors are correct that current models of T2DM do not fully recapitulate the disease progression, and symptoms observed in patients. However, I do feel that the author's claim that this model is able to "reproduce the complex metabolic disturbances associated with early-stage type 2 diabetes in humans" is overstated given the relatively limited characterisation work performed in this rodent model (compared to the totality of human T2DM characterisation and investigation). Additionally, attempting to recreate the full spectra of symptoms and co-morbidities observed in human disease, is for me, of questionable value given recent work demonstrating that T2D is a highly heterogeneous disease (e.g. <https://doi.org/10.2337/dbi20-0001>). Thus my main comments are related to tempering of certain claims made by the authors.

Authors response: *we thank reviewer 2 for the positive comments provided regarding our manuscript. We agree that some of the statements made were perhaps too broad in nature and have revised the manuscript based on the individual comments provided below.*

Comments

1. Title: "Development and characterisation of a rat model that exhibits both metabolic 2 dysfunction and neurodegeneration seen in humans with type 2 diabetes" I don't think that the manner of metabolic dysfunction and neurodegeneration is precisely observed in humans. Therefore the authors should remove "seen in humans with type 2 diabetes" from the title.

Authors response: *we agree, and we have removed the word human from the title to ensure we are more accurately representing the nature of the work in the manuscript. We hope this is enough of a change to satisfy the reviewers concerns regarding the comparisons/conclusions regarding human type 2 diabetes.*

Changes to manuscript: *the title of the manuscript has been changed to "Development and characterisation of a rat model that exhibits both metabolic dysfunction and neurodegeneration seen in type 2 diabetes"*

2. "we wanted to understand whether these animals developed the classic neurological complications of type 2 diabetes such as peripheral neuropathy - a major cause of morbidity that effects approximately 50% of people with type 2 diabetes" The prevalence of neuropathy in T2DM appears to vary in the literature e.g. Feldman et al., 2019 suggests 8-51% prevalence. Perhaps the authors could briefly discuss the range of reported prevalence, and any relevance this might have to their model.

Authors response: *the reviewer is correct and the use of 50% is not an accurate description of the prevalence of diabetic neuropathy in the human population.*

Changes to manuscript: *we have modified the language in this part of the manuscript (lines 575-580) to state the following and included another statement in the limitations section regarding this.*

Lines 575-580: "Next, we wanted to understand whether these animals developed the classic neurological complications of type 2 diabetes such as peripheral neuropathy – a major cause of

morbidity that effects as many as 50% of people with type 2 diabetes across the time-course of their disease."

Lines 635-646: "Another limitation of the work is the timeframe during which the type 2 diabetic animals develop signs of peripheral neuropathy. While our animals develop signs of hypersensitive neuropathy after 9-10 weeks, in humans with type 2 diabetes peripheral neuropathy typically takes 5-10 years to clinically manifest. The difference in development and progression of peripheral neuropathy between our animal model and that observed in humans may be due to the level and quality of gluco-regulatory control. Humans are typically prescribed glucose lowering medications such as metformin which may delay development of neurodegeneration. In contrast, our animals are not medicated and have chronic hyperglycaemia which may drive accelerated peripheral neurodegeneration. Therefore, although our model recapitulates the neurodegenerative complications of type 2 diabetes, the timeframe during which this occurs does not represent the human course of the disease."

3. The authors appear to highlight certain aspects of their model's timeline of symptom progression when it suits, and 'play-down' other aspects when it doesn't. E.g. altered hind paw withdrawal threshold was observed already at week 9, whereas, in the human condition neuropathic symptoms are progressive and occur secondary, often over a long period of time, to poor gluco-regulatory control. From Feldman et al., 2019 : "Indeed, the prevalence of diabetic neuropathy increased from 8% to 42% in patients with T2DM when patients were monitored for 10 years". The authors should be clear that the timeline of development of peripheral neuropathy in their model does not recapitulate disease progression in many people with T2DM.

Authors response: *this is a related point to the previous comment. Again, we agree with the reviewer and our language regarding this point was not clear enough.*

Changes to manuscript: *we have modified language in the limitations section as noted in the previous comment (lines 635-646 – see above) to highlight the difference in the time-course of neuropathy development in humans vs rodents and provided some discussion to further elaborate on this point.*

4. Given the comments above, it is clear that this concluding sentence is overstated and should be revised: "Therefore, this model recapitulates the complex and multiple metabolic disturbances seen in people with type 2 diabetes that will improve our understanding of the mechanisms that lead to neurodegeneration as well as providing a powerful platform for testing new therapeutics to limit nervous system damage in type 2 diabetes."

Authors response: *this is in line with the other comments provided and we have revised the final sentence of the manuscript and the final line of the abstract accordingly. In a number of other sentences in the manuscript, we have also replaced 'human type 2 diabetes' and with just 'type 2 diabetes' to limit some of the overstated outcomes from the new animal model as it relates to the human condition.*

Changes to manuscript: *we have modified language in a number of sections of the manuscript as below:*

Line 42-44 (key points): "Over four months, we comprehensively characterised these animals and confirmed they develop sustained metabolic dysfunction and progressive peripheral neurodegeneration as seen in type 2 diabetes"

Lines 63-65 (abstract): "Thus, this model recapitulates many of the complex metabolic disturbances seen in type 2 diabetes and provides an excellent platform to investigate the pathophysiological mechanisms that lead to diabetic complications like peripheral neuropathy."

Lines 655-659 (discussion): "Therefore, this model recapitulates many of the key metabolic disturbances seen in type 2 diabetes and provides an excellent platform for testing new therapeutics to prevent or reverse complications of type 2 diabetes such as peripheral neuropathy."

Dear Dr Premilovac,

Re: JP-TFP-2021-282454R1-A "Development and characterisation of a rat model that exhibits both metabolic dysfunction and neurodegeneration seen in type 2 diabetes" by Katherine Southam, Chantal de Sousa, Abraham Daniel, Bruce Taylor, Lisa Foa, and Dino Premilovac

Thank you for submitting your manuscript to The Journal of Physiology. It has been assessed by a Reviewing Editor and by 2 expert referees and I am pleased to tell you that it is considered to be acceptable for publication following satisfactory revision.

The reports are copied at the end of this email. Please address all of the points and incorporate all requested revisions, or explain in your Response to Referees why a change has not been made.

NEW POLICY: In order to improve the transparency of its peer review process The Journal of Physiology publishes online as supporting information the peer review history of all articles accepted for publication. Readers will have access to decision letters, including all Editors' comments and referee reports, for each version of the manuscript and any author responses to peer review comments. Referees can decide whether or not they wish to be named on the peer review history document.

Authors are asked to use The Journal's premium BioRender (<https://biorender.com/>) account to create/redrawn their Abstract Figures. Information on how to access The Journal's premium BioRender account is here: <https://physoc.onlinelibrary.wiley.com/journal/14697793/biorender-access> and authors are expected to use this service. This will enable Authors to download high-resolution versions of their figures.

I hope you will find the comments helpful and have no difficulty returning revisions within 4 weeks.

If you need to check to make sure that your Methods section conforms to the principles of UK regulations, you may wish to refer to Grundy (2015):

Grundy (2015) J. Physiol. 2015 Jun 15;593(12):2547-9 <https://doi.org/10.1113/JP270818>

Your revised manuscript should be submitted online using the links in Author Tasks Link Not Available. This link is to the Corresponding Author's own account, if this will cause any problems when submitting the revised version please contact us.

The image files from the previous version are retained on the system. Please ensure you replace or remove any files that have been revised.

REVISION CHECKLIST:

- Summary data must be reported as mean {plus minus} SD or 95% confidence interval
- All table and figure legends with summary data must include the statistical test used in the table/figure and sample size
- Figures with summary data bars must include individual data points, or box whisker plots when $n < 30$.
- Article file, including any tables and figure legends, must be in an editable format (eg Word)
- Abstract figure file (see above)
- Statistical Summary Document
- Upload each figure as a separate high quality file
- Upload a full Response to Referees, including a response to any Senior and Reviewing Editor Comments;
- Upload a copy of the manuscript with the changes highlighted.

- A potential 'Cover Art' file for consideration as the Issue's cover image;
- Appropriate Supporting Information (Video, audio or data set https://jp.msubmit.net/cgi-bin/main.plex?form_type=display_requirements#supp).

To create your 'Response to Referees' copy all the reports, including any comments from the Senior and Reviewing Editors, into a Word, or similar, file and respond to each point in colour or CAPITALS and upload this when you submit your revision.

I look forward to receiving your revised submission.

If you have any queries please reply to this email and staff will be pleased to assist.

Yours sincerely,

Professor Kim E. Barrett
Editor-in-Chief
The Journal of Physiology
<https://jp.msubmit.net>
<http://jp.physoc.org>
The Physiological Society
Hodgkin Huxley House
30 Farringdon Lane
London, EC1R 3AW
UK
<http://www.physoc.org>
<http://journals.physoc.org>

REQUIRED ITEMS:

-Papers must comply with the Statistics Policy https://jp.msubmit.net/cgi-bin/main.plex?form_type=display_requirements#statistics

In summary:

-If $n \leq 30$, all data points must be plotted in the figure in a way that reveals their range and distribution. A bar graph with data points overlaid, a box and whisker plot or a violin plot (preferably with data points included) are acceptable formats.

-If $n > 30$, then the entire raw dataset must be made available either as supporting information, or hosted on a not-for-profit repository e.g. FigShare, with access details provided in the manuscript.

- n clearly defined (e.g. x cells from y slices in z animals) in the Methods. Authors should be mindful of pseudoreplication.

-All relevant n values must be clearly stated in the main text, figures and tables, and the Statistical Summary Document (required upon revision)

-The most appropriate summary statistic (e.g. mean or median and standard deviation) must be used. Standard Error of the Mean (SEM) alone is not permitted.

-Exact p values must be stated. Authors must not use 'greater than' or 'less than'. Exact p values must be stated to three significant figures even when 'no statistical significance' is claimed.

-Statistics Summary Document completed appropriately upon revision

EDITOR COMMENTS

Reviewing Editor:

Thank you for addressing the comments of the reviewers. The revised manuscript has been provisionally accepted for publication to the Journal of Physiology.

Could you please address the following two comments in the updated version of the manuscript?

1) In the statistical analysis section, you mentioned that you performed a Kruskal-Wallis test to check for normality. Could you please verify that this is correct?

2) The colors used for the figures (particularly red and green) may not be visible to colorblind people. I suggest selecting alternative colors that can be visible to colorblind readers.

Senior Editor:

The manuscript is not fully compliant with our statistical policies. There are numerous places where $p < 0.001$ is used, which is not allowed - precise p values must be given unless $p < 0.00001$. Also, raw data are not uniformly supplied.

REFEREE COMMENTS

Referee #1:

The authors have responded adequately to the reviewer comments.

Referee #2:

I thank the authors for carefully responding to my comments and congratulate them on an excellent manuscript.

END OF COMMENTS

1st Confidential Review

16-Dec-2021

Re: JP-TFP-2021-282454 "Development and characterisation of a rat model that exhibits both metabolic dysfunction and neurodegeneration seen in humans with type 2 diabetes" by Katherine Southam, Chantal de Sousa, Abraham Daniel, Bruce Taylor, Lisa Foa, and Dino Premilovac

Dear Professor Barrett,

The authors would like to thank the reviewers and editors for provisionally accepting our manuscript for publication in the Journal of Physiology. Below is a point-by-point summary of the changes that have been made to the manuscript based on the comments provided.

Reviewing Editor:

Thank you for addressing the comments of the reviewers. The revised manuscript has been provisionally accepted for publication to the Journal of Physiology.

Could you please address the following two comments in the updated version of the manuscript?

1) In the statistical analysis section, you mentioned that you performed a Kruskal-Wallis test to check for normality. Could you please verify that this is correct?

Authors response:

We thank the reviewing editor for picking up this typographical error. Normality was assessed using the Shapiro-Wilk and not the Kruskal-Wallis test. This typographical error in the statistics section on page 12 has been corrected as below.

Changes to manuscript:

“Where data was normally distributed, as determined by Shapiro-Wilk normality test,...”

2) The colors used for the figures (particularly red and green) may not be visible to colorblind people. I suggest selecting alternative colors that can be visible to colorblind readers.

Authors response:

We have modified the colour palate where appropriate and replaced the colour red within the data panels/symbols. We have not altered the red/green colour palate in the immunohistochemical images (figs 8 and 9) because these are the true fluorophores and thus best represent the data.

Changes to manuscript:

All figure panels have had red symbols replaced with a lighter shade of purple to help with accessibility. All other colours remain the same.

Senior Editor:

The manuscript is not fully compliant with our statistical policies. There are numerous places where

$p < 0.001$ is used, which is not allowed - precise p values must be given unless $p < 0.00001$. Also, raw data are not uniformly supplied.

Authors response:

All statistics are now reported to 4 decimal places both in text and within the figures. As per the journals policy, we have only used ' $p <$ ' when referring to less than 0.0001.

Changes to manuscript:

All statistics in the written results section (pages 13-22) are now reported to four decimal places. Similarly, all figures now also show statistics to four decimal places.

REFEREE COMMENTS

Referee #1:

The authors have responded adequately to the reviewer comments.

Referee #2:

I thank the authors for carefully responding to my comments and congratulate them on an excellent manuscript.

Dear Dr Premilovac,

Re: JP-TFP-2022-282454R2 "Development and characterisation of a rat model that exhibits both metabolic dysfunction and neurodegeneration seen in type 2 diabetes" by Katherine Southam, Chantal de Sousa, Abraham Daniel, Bruce Taylor, Lisa Foa, and Dino Premilovac

I am pleased to tell you that your paper has been accepted for publication in The Journal of Physiology, subject to any modifications to the text and/or satisfactory clarification of the Methods section that may be required by the Journal Office to conform to House rules.

NEW POLICY: In order to improve the transparency of its peer review process The Journal of Physiology publishes online as supporting information the peer review history of all articles accepted for publication. Readers will have access to decision letters, including all Editors' comments and referee reports, for each version of the manuscript and any author responses to peer review comments. Referees can decide whether or not they wish to be named on the peer review history document.

The last Word version of the paper submitted will be used by the Production Editors to prepare your proof. When this is ready you will receive an email containing a link to Wiley's Online Proofing System. The proof should be checked and corrected as quickly as possible.

The accepted version of the manuscript is the version that will be published online ahead of the copy edited and typeset version being made available. Authors should note that it is too late at this point to offer corrections prior to proofing. Major corrections at proof stage, such as changes to figures, will be referred to the Reviewing Editor for approval before they can be incorporated. Only minor changes, such as to style and consistency, should be made a proof stage. Changes that need to be made after proof stage will usually require a formal correction notice.

All queries at proof stage should be sent to TJP@wiley.com

Are you on Twitter? Once your paper is online, why not share your achievement with your followers. Please tag The Journal (@jphysiol) in any tweets and we will share your accepted paper with our 22,000 plus followers!

Yours sincerely,

Professor Kim E. Barrett
Editor-in-Chief
The Journal of Physiology
<https://jp.msubmit.net>
<http://jp.physoc.org>
The Physiological Society
Hodgkin Huxley House
30 Farringdon Lane
London, EC1R 3AW
UK
<http://www.physoc.org>
<http://journals.physoc.org>

P.S. - You can help your research get the attention it deserves! Check out Wiley's free Promotion Guide for best-practice recommendations for promoting your work at www.wileyauthors.com/eeo/guide. And learn more about Wiley Editing Services which offers professional video, design, and writing services to create shareable video abstracts, infographics, conference posters, lay summaries, and research news stories for your research at www.wileyauthors.com/eeo/promotion.

*** IMPORTANT NOTICE ABOUT OPEN ACCESS ***

Information about Open Access policies can be found here <https://physoc.onlinelibrary.wiley.com/hub/access-policies>

To assist authors whose funding agencies mandate public access to published research findings sooner than 12 months after publication The Journal of Physiology allows authors to pay an open access (OA) fee to have their papers made freely available immediately on publication.

You will receive an email from Wiley with details on how to register or log-in to Wiley Authors Services where you will be able to place an OnlineOpen order.

You can check if your funder or institution has a Wiley Open Access Account here <https://authorservices.wiley.com/author-resources/Journal-Authors/licensing-and-open-access/open-access/author-compliance-tool.html>

Your article will be made Open Access upon publication, or as soon as payment is received.

If you wish to put your paper on an OA website such as PMC or UKPMC or your institutional repository within 12 months of publication you must pay the open access fee, which covers the cost of publication.

OnlineOpen articles are deposited in PubMed Central (PMC) and PMC mirror sites. Authors of OnlineOpen articles are permitted to post the final, published PDF of their article on a website, institutional repository, or other free public server, immediately on publication.

Note to NIH-funded authors: The Journal of Physiology is published on PMC 12 months after publication, NIH-funded authors DO NOT NEED to pay to publish and DO NOT NEED to post their accepted papers on PMC.

EDITOR COMMENTS

Reviewing Editor:

Congratulations!

2nd Confidential Review

23-Jan-2022